# Physical Modulation to the Biological Productivity in the Summer Vietnam Upwelling System

Wenfang Lu[1,2,3,4], Lie-Yauw Oey[5,6], Enhui Liao[6], Wei Zhuang[2,4], Xiao-Hai Yan[3,4], Yuwu Jiang[2,4]

[1]Key Laboratory of Spatial Data Mining and Information Sharing of Ministry of Education, & National Engineering Research Centre of Geo-spatial Information Technology, Fuzhou University, Fuzhou, 350116, China
[2]State Key Laboratory of Marine Environmental Science, College of Ocean and Earth Sciences, Xiamen University, Xiamen, 361102, China
[3]Center for Remote Sensing, College of Earth, Ocean and Environment, University of Delaware, Newark, DE, USA
[4]Joint Institute for Coastal Research and Management, University of Delaware/Xiamen University, USA/China
[5]National Central University, Jhongli City, Taoyuan County, Taiwan
[6]Princeton University, Princeton, New Jersey, USA

*Correspondence to*: Yuwu Jiang (ywjiang@xmu.edu.cn)

**Abstract.** Biological productivity in the summer Vietnam boundary upwelling system in the western South China Sea, as in many coastal upwelling systems, is strongly modulated by wind. However, the role of ocean circulation and mesoscale eddies has not been elucidated. Here we show a close spatiotemporal covariability between primary production and kinetic energy. High productivity is associated with high kinetic energy, which accounts for ~15% of the production variability. Results from a physical-biological coupled model reveal that the elevated kinetic energy is linked to the strength of the current separation from the coast. In the low production scenario, the circulation is not only weaker but also shows weak separation. In the higher production case, the separated current forms an eastward jet into the interior South China Sea, and the associated southern recirculation traps nutrient and favors productivity. When separation is absent, the model shows weakened circulation and eddy activity, with ~21% less nitrate inventory and ~16% weaker primary productivity.

## 1    Introduction

The South China Sea (SCS) is a large semi-enclosed marginal sea located in the western Pacific Ocean (Fig. 1a). It is bordered by extensive continental shelves along the southern coast of China and northeastern Vietnam, and the Sunda Shelf south of Vietnam (Fig.1). It has a deep interior basin which can be as deep as 5000 m (Liu et al., 2010; Wong et al., 2007). The SCS is predominantly controlled by the East Asian Monsoon. The wind is southwesterly from June to September, and northeasterly from November to March (Liu et al., 2002). Because of efficient biological production,

the interior SCS has a low nutrient concentration in the euphotic zone, displaying an oligotrophic condition (Wong et al., 2007).

Coastal upwelling is one of the most important processes for ocean productivity and fishery (Bakun, 1996; Cushing, 1969; Mittelstaedt, 1986). During southwesterly monsoon, upwelling-favorable wind prevails along the southern coast of Vietnam over the complex topography (Fig. 1b). The offshore Ekman transport drives surface divergence and

results in coastal upwelling of cold and nutrient-rich subsurface water. We refer to this region of interest as the Vietnam Boundary Upwelling System (VBUS). The VBUS is centered near ~109° E between 14° N and 17° N along the coast (Loisel et al., 2017). Upwelling in VBUS was confirmed by cruise  (Dippner et al., 2007) and remote sensing observations (Kuo et al., 2000).

In VBUS, the upwelling intensity is governed by the strength of the alongshore monsoon wind, as in other coastal

upwelling systems such as the coastal upwelling systems of California and mid-Atlantic Bight (Gruber et al., 2011). The VBUS upwelling strength is intense, and can result in surface cooling of 3~5 °C and an associated cold filament length of ~500 km (Kuo et al., 2004). The VBUS is modulated by different climatic variations, such as the El Niño and Southern Oscillation (Dippner et al., 2007; Hein et al., 2013; Xie et al., 2003), the Indian Ocean Dipole (Liu et al., 2012; Xie et al., 2009), and the Madden-Julian Oscillation (Isoguchi and Kawamura, 2006; Liu et al., 2012).

Previous studies suggested that both the wind-induced upwelling and the local circulation could influence the nutrient balance and ecosystem in the VBUS. The El Niño variability is an important controlling factor of the VBUS by modulating the summer monsoon (Chai et al., 2009; Kuo and Ho, 2004). During post-El Niño summer, the weakened southwesterly wind leads to weak upwelling and reduced upward nutrient flux (Xie et al., 2003). In addition, Hein et al. (2013) proposed instead that productivity was controlled by lateral transport of nitrate in the VBUS. Liu et

al. (2002) also highlighted the role of coastal jet located to the south of Vietnam coast. They mentioned that jet-induced upwelling was responsible for the nutrient influx. Xie et al. (2003) mentioned the role of the offshore jet and resultant quasi-stationary eddy (i.e., the "recirculation" hereafter) in transporting the highly productive water. However, the contribution from recirculation has not been directly quantified and compared with that directly from the upwelling, which motivates us to revisit the VBUS ecosystem and its connection with circulation.

Early hydrodynamic observations revealed a northeastward coastal current over the southern shelf of Vietnam (Wyrtki, 1961). The current separates from the coast and flows offshore at about 11°N (Xu et al., 1982). Xie et al. (2003) ascribed the jet separation to the strong wind jet off Vietnam due to the orographic steering of the north-south running mountains. Using an idealized reduced gravity model, Wang et al. (2006) highlighted vorticity input by wind-

stress curl and vorticity advection by the basin circulation. Gan and Qu (2008) found that the separation was associated with an adverse pressure gradient induced by the topographic effects.

The separated jet produces cooling and results in biannual SST variation in the SCS (Xie et al., 2003). The offshore jet also appears to advect water with high chlorophyll to the interior of the central SCS (Chen et al., 2014; Loisel et al., 2017; Tang et al., 2004). While the importance of the local current system to the VBUS biogeochemical system has been noted in some previous studies (Dippner et al., 2007; Kuo et al., 2004; Liu et al., 2012; Xie et al., 2003), the detailed processes are unclear. To what extent is the ecosystem in VBUS modulated by local circulation? Does the circulation show any distinction in the high/low production cases? How does the recirculation modulate productivity? How much does the local circulation contribute to production? Studying biological production and its coupling with physical processes in the VBUS will help to answer these questions and further improve the understanding of boundary upwelling system. Such a study will also shed light on the ecosystem dynamics in the SCS as an oligotrophic marginal sea. Here we analyze the complex dynamics of the VBUS using a physical-biological coupled numerical model system, as well as remote sensing data and *in situ* observations.

This paper is organized as follows. In Sect. 2, the model configuration, numerical experiments, observed data, and statistical method used in this study are described. In Sect. 3, we analyze the remote sensing data and validate the model. Model results from both the standard run and the sensitivity experiment are presented. In Sect. 4, the dynamical processes are analyzed. Conclusions are given in Sect. 5.

## 2 Model, Data and Methods

### 2.1 Data

The surface wind vectors were from the Cross-Calibrated Multi-Platform (CCMP) gridded data. This is a 25-year, six-hourly, $1/4° \times 1/4°$ resolution product fused from several microwave radiometers and scatterometers using a variational analysis method (Atlas et al., 2011). Monthly Moderate Resolution Imaging Spectroradiometer (MODIS) Aqua level-3 chlorophyll (4 km resolution) was obtained from the NASA Distributed Active Archive Center. The estimated monthly vertical-integrated net primary production (NPP) was derived from MODIS chlorophyll data via the standard chlorophyll-based Vertically Generalized Production Model (VGPM) algorithm (Behrenfeld and Falkowski, 1997)). The VGPM NPP product had a resolution of 1/10 degree, covering the period from 2004 to present. Gridded monthly-mean Absolute Dynamic Topography (ADT) with respect to the geoid at $1/4°$ resolution was acquired. The $1/4°$ Optimum Interpolation Sea Surface Temperature (OISST, also known as Reynolds 0.25v2) was constructed by combining the Advanced Very High Resolution Radiometer satellite and other observation data (Banzon et al., 2016). Atmospheric forcing including downward shortwave radiation, downward longwave radiation, air temperature, air pressure, precipitation rate and relative humidity were acquired from the National Centers for Environmental Prediction Reanalysis (Kalnay et al., 1996). The river runoff data (see Sect. 2.3 below) was adopted from Dai et al. (2009), which contains observation-based monthly freshwater runoff of major rivers of the world. The climatology of temperature, salinity, nutrients, and dissolved organic matter was adopted from the World Ocean Atlas (WOA, 2013

version). *In situ* observed nitrate and chlorophyll profiles from the western SCS stations (Fig. 1b) were used, as detailed in Jiao et al. (2014).

**2.2 Methods**

2.2.1 Upwelling Intensity (UI) and Kinetic Energy (KE)

We use the upwelling intensity (UI) or the "Bakun index" (Bakun, 1973) as a proxy to measure the strength of upwelling (Chen et al., 2012; Gruber et al., 2011), following the classical paper of Ekman (1905):

$$\mathbf{UI} = \frac{\tau_y}{\rho_0 f} = \frac{\rho_a C_D U_y |U_y|}{\rho_0 f}. \tag{1}$$

Here, $\tau_y$ is the along-shore component of wind stress, $f$ is the Coriolis parameter, $\rho_0$ is seawater density (constant, 1025 kg m$^{-3}$), $\rho_a$ is the air density (constant, 1.2 kg m$^{-3}$), $C_D$ is the drag coefficient (constant, $1.3 \times 10^{-3}$) and $U_y$ is the alongshore wind speed. The CCMP data with full temporal and spatial coverage close to the coastline is used for the wind speed.

     The kinetic energy (KE) of the near-surface current is used as an indicator of the circulation intensity. The near-
surface current is calculated from absolute dynamic topography (ADT) using the geostrophic balance. The KE then equals:

$$\mathrm{KE} = \frac{1}{2}\left(u_g^2 + v_g^2\right) = \frac{1}{2\rho_0^2 f^2}\left[\left(\frac{\partial \mathrm{ADT}}{\partial x}\right)^2 + \left(\frac{\partial \mathrm{ADT}}{\partial y}\right)^2\right], \tag{2}$$

2.2.2 Multivariable Linear Regression

Monthly net primary production (NPP) from VGPM (Vertically Generalized Production Model; see Sect. 2.1 Data)
was used to estimate biological productivity. A multivariable linear regression analysis was conducted to examine the statistical relations among NPP, UI, and KE:

$$NPP = b_1\ UI + b_2\ KE + b_3, \tag{3}$$

where $b_1$, $b_2$, and $b_3$ are the estimated parameters of the regression. Data in the summer months (MJJAS) were used since the monsoon wind during this period is upwelling-favorable. The focus region of this study is not confined to
the "actual" upwelling strip of ~40 km wide as indicated by the first baroclinic Rossby radius of deformation (Dippner et al., 2007; Voss et al., 2006), but extending to a broader offshore region of ~three-degree wide. Therefore, we averaged NPP and KE over the ocean region enclosed by the magenta contour off the coast of Vietnam (Fig. 2b). Only

the summertime data in the overlapping period from 2004 to 2012 were analyzed. Contributions from SST, day length, and the photosynthetically active radiation were implicitly considered in the VGPM (Behrenfeld and Falkowski, 1997).

## 2.3 Model Description

We use a three-dimensional general circulation model based on the Regional Ocean Model System (ROMS). ROMS is a free-surface and hydrostatic ocean model. It solves the Reynolds-averaged Navier-Stokes equations on terrain-following coordinates (Shchepetkin and McWilliams, 2005). The model is used in the operational Taiwan Strait Nowcast\Forecast system (TFOR), which successfully provides multi-purpose ocean forecasts (Jiang et al., 2011; Liao et al., 2013; Lin et al., 2016; Lu et al., 2017; Lu et al., 2015; Wang et al., 2013). In this study, the model grid is modified to cover the whole SCS domain and part of the North-Western Pacific with a grid resolution of 1/10 degree (Fig. 1a). The number of grid nodes in *x*- and *y*-direction are 382 and 500, respectively. In the vertical, 25 $\sigma$-levels is used with a grid size of ~2 m on average near the surface to resolve the surface boundary layer. Following the bulk formulation scheme (Liu et al., 1979), daily atmospheric fluxes (detailed in Sect. 2.1) are applied at the surface. The wind vectors are from the CCMP wind. The vertical turbulent mixing uses a K-profile parameterization (KPP) scheme (Large et al., 1994) which was successfully applied in a one-dimensional vertical mixing model in the SCS (Lu et al., 2017). The KPP scheme estimates eddy viscosity within the boundary layer as the production of the boundary layer depth, a turbulent velocity scale, and a dimensionless third-order polynomial shape function. Beyond the surface boundary layer, KPP scheme includes vertical mixing collectively contributed by shear mixing, double diffusive process and internal waves. Biharmonic horizontal mixing scheme (Griffies and Hallberg, 2000) with a reference viscosity of $2.7 \times 10^{10}$ $m^4$ $s^{-1}$ is applied, following the value of Bryan et al. (2007) used in a circulation model with the same horizontal resolution. Climatological river discharges from the Mekong River and other major rivers are included as point sources.

The biogeochemical module is the Carbon, Silicon, Nitrogen Ecosystem (CoSINE) model (Xiu and Chai, 2014), which consists of 31 state variables, including four nutrients [nitrate ($NO_3$), ammonium ($NH_4$), silicate, and phosphate], three phytoplankton functional groups (representing picoplankton, diatoms and coccolithophorids), two zooplankton classes (i.e., microzooplankton and mesozooplankton), four detritus pools (particulate organic nitrogen/carbon, particulate inorganic carbon, and biogenic silica), four dissolved organic matters (labile and semi-labile pools for both carbon and nitrate), and bacteria. Other planktonic groups can be important in the ecosystem of SCS in some condition (Bombar et al., 2011; Doan-Nhu et al., 2010; Loick-Wilde et al., 2017). Surely, adding more planktonic groups would better depict the complex relationship in the ecosystem. However, considering the functional groups chosen here were dominating species widely observed in the SCS [e.g., Ning et al. (2004)], adding more planktonic species is unlikely to radically change the spatiotemporal variation of the modeled ecosystem. To keep the ecosystem model simple and computationally affordable, these groups are not considered in the CoSINE model. The CoSINE model was well applied in various studies of the SCS. Liu and Chai (2009) investigated the seasonal and interannual variability of the primary productivity of the SCS at a basin scale. The modeled structure (e.g., phytoplankton community) and function

(e.g., biological pump) of the ecosystem appeared to respond to both climatic variations (Ma et al., 2013, 2014) and mesoscale eddies (Guo et al., 2015), both well-captured by CoSINE. By taking a SCS average, the modeled NPP time-series showed a strong correlation (R=0.84) when compared with satellite-derived production (Ma et al., 2014). Details of the CoSINE results in SCS can also be referred to Lu et al. (2018).


The physical modeled was initialized from a resting state with temperature and salinity specified using the WOA climatology. The initial distribution of the nutrients and dissolved organic matter was also interpolated from the WOA climatological data. Our previous studies related with ecosystem modeling in the China Seas (Lu et al., 2015; Wang et al., 2016; Wang et al., 2013) suggested that the ecosystem module was more sensitive to the initial value of nutrients and dissolved organic matters. For other variables (i.e., detritus and planktons), the model converged to similar states


even when these variables were initialized differently. Hence, for these ecosystem variables, small values were assigned as in Table S1. After spinning up for 13 years with climatological forcing, the model was restarted with the ecosystem module driven by interannually-varying CCMP wind and atmospheric surface forcing from 2002 to 2011. The model outputs from 2005 to 2011 are analyzed.

## 2.4 Sensitivity Experiment

To quantify the contribution to the ecosystem from the recirculation, we seek to control the formation of the recirculation while maintaining the larger basin-scale circulation. For the Vietnam boundary upwelling system, since nonlinear advection is important to the separation of the coastal jet and thus the formation of the anticyclone (Gan and Qu, 2008; Wang et al., 2006), which is familiar in the Gulf Stream separation problem (Haidvogel et al., 1992;


Marshall and Tansley, 2001), an experiment without the nonlinear advection terms in the momentum equations was conducted [following, e.g., Gruber et al. (2011)]. It should be noted that the advection terms in the tracer equations are retained for transport of active and passive tracers (i.e., ecosystem variables). Hereafter, this experiment will be referred to as NO_ADV run.

### 3    Results

In this section, we first analyze the satellite-based observational data, focusing on the spatiotemporal covariance of wind, circulation, and biological production. After accessing the model performance against observation, we then describe and discuss the model results.

## 3.1 Spatio-temporal Analysis of Observation Data

Figure 2 shows the mean (Fig. 2a) and standard deviation (Fig. 2b) of surface chlorophyll overlaid with contours of

mean ADT and KE respectively. In summer, the surface chlorophyll has a low concentration of <0.1 mg m$^{-3}$ in the central SCS basin. By contrast, the chlorophyll is more than fivefold (>0.5 mg m$^{-3}$) along the southern Vietnamese coast. The high chlorophyll water appears to extend offshore following the coastal jet to the interior SCS. The jet overshoots after separating from the coast and bifurcates into a northeastward current and a quasi-stationary anti-

cyclonic eddy (Fig. 2a). Centered at ~11° N near the tip of Vietnamese coast, high KE (>1.0 m$^2$ s$^{-2}$) appears near the coast. The high variability of chlorophyll coincides with KE into the interior SCS, implying the contribution from the jet (Fig. 2b).

The box-averaged (magenta box in Fig. 2b) time-series of monthly UI, KE and NPP are shown in Fig. 3a-c; they show seasonal and interannual variations. KE and NPP both present biannually signals in most years, i.e., peaks in summer and winter, as well as complex non-seasonal signals. Unsurprisingly, UI dominates about half (R$^2$=0.4548 for UI solely, p<0.01) of the total variability in NPP, which is consistent with studies in other wind-driven upwelling systems (Gruber et al., 2011), and with the previous studies of the VBUS (Bombar et al., 2010; Voss et al., 2006). The correlation between KE and NPP is even higher (R$^2$=0.4930, p<0.01). Moreover, although KE could be dependent on the wind, the R$^2$ between KE and UI is 0.3240, suggesting that a large part (~68%) of the variation in KE is unexplained by the uniform alongshore wind. There are clear positive contributions to the biological production from both UI and KE. When KE and UI are considered concurrently, additional ~15% of the variability in NPP is explained (R$^2$=0.6046, p<0.01).

To further illustrate the modulation in the ecosystem by circulation, the flow pattern in high-NPP-anomaly (HNA) and low-NPP-anomaly (LNA) were composited according to the de-seasonalized NPP anomaly (Fig. 3c). The seasonal signal was firstly removed from the summertime NPP, yielding the de-seasonalized NPP anomaly. The thresholds for HNA and LNA are defined as (above) 75% and (below) 25% percentile of the NPP anomaly, respectively. Different thresholds (60% and 70%) were also tested and very similar results can be seen. The velocity and direction for LNA, HNA and the normal state (i.e., neither LNA nor HNA) are respectively depicted in Fig. 4a-c, as well as the ADT difference between HNA and LNA (Fig. 4d). A student t-test suggests that the three circulation patterns are significantly (*p*<0.01) different. In contrast to the familiar separation and offshore jet pattern (Fig. 2a and Fig. 4c), the LNA circulation tends to flow along the coast without separation (Fig. 4a). On the other hand, the HNA circulation (Fig.4b) shows a clear separated jet and anticyclonic recirculating pattern south of the jet near 8.5° N, similar to the pattern seen during normal years (Fig. 4c); the flow speed is ~20% stronger than that of the normal state. Near the separation point, the HNA jet is more dissipated and slightly weakened compared with the LNA coastal jet. The KE averaged within the magenta box (Fig.2b) during the HNA state is 0.0827 m$^2$ s$^{-2}$, which is ~65% larger than that during the LNA state (0.0502 m$^2$ s$^{-2}$). The difference in flow patterns is consistent with a dipolar ADT difference, by which a westward (inverse to the jet) pressure gradient force anomaly is imparted to the flow (Fig. 4d), which is responsible for the jet separation process (Batchelor, 1967; Gan and Qu, 2008). In order to examine the recirculation's role in the ecosystem, we now use the model to address the physical-biogeochemical coupling.

### 3.2 Model Validation

In Fig. 5, simulated SST and NPP are compared with observations. The model reproduces reasonably well the observed patterns of SST and NPP. In particular, the model captures the cross-shore SST gradient. The cold filament that overshoots from the coast to the interior of SCS is also clearly reproduced by the model. However, the modeled

SST shows a systematic cold bias of ~1 °C, and the modeled NPP does not simulate well the extremely high values (>1000 mg C m$^{-2}$ d$^{-1}$) along the Vietnamese coast. This may in part be attributed to overestimation of retrieved NPP near the coast (Loisel et al., 2017). Off the coast, the model simulates well the cross-shore gradient of productivity. The gradient is generally high in areas influenced by the jet. In the coupled model, while it is true that SST affects NPP through, for example, changes in the vertical stratification of the water column, both SST and NPP strongly depend on circulation (e.g. upwelling and/or downwelling), and in our case on the flow separation and KE also. In turn, the circulation is dominated by changes in the upper-layer depth (as diagnosed through the SSH) and the horizontal gradients of SSH, and is much less dependent on the gradients of SST. Thus, the co-variation between the SST and ecosystem is largely controlled by the circulation. The dominant ecosystem response is the separation and non-separation contrast, which is captured well by the model (comparing Fig. 4 and Fig. 8).

Time series of modeled SST, surface KE and NPP, averaged over the magenta contour region (Fig.2b), are compared with observations in Fig. 6. Due in part to the realistic surface forcing and high resolution used, our model can reproduce the physical and biological variables in the VBUS. The seasonal cycles in all three quantities agree reasonably well with the observations. At interannual time scales, during the 2010 El Niño event, for example, monsoon was weaker (Fig. 3a), SST was warmer, and the KE was reduced. These features are simulated well, although the production drawdown is slightly weaker than the observation and the simulated SST under-estimates amplitude of the observed SST annual cycle by ~1.0 °C. For the surface current and productivity, our model shows excessive KE and insufficient production during winter, but the model-observation discrepancy is less notable in other seasons. The overestimated KE is partially contributed by the Ekman component in the modeled surface current. Nevertheless, we can conclude that our model reasonably reproduced the temporal variability in the VBUS.

In addition, vertical profiles of the simulated NO$_3$ and chlorophyll, as two fundamental components of the marine ecosystem, are compared with observations (Fig. 7). The modeled NO$_3$ generally reflects the oligotrophic condition near the surface and the nutricline approximately at 50 m. Below the nutricline, the NO$_3$ profile shows moderate vertical gradient to the deep. The simulated NO$_3$ profile matches the observations remarkably well. For the chlorophyll, our model well simulates the concentration, not only at the surface but also in the deep layer. A subsurface chlorophyll maxima appears at ~35 m, which is somewhat shallower than that in the observation (50 m). Except for model uncertainty, this discrepancy may also be related to the undersampling in observed profiles (no water samples between 25 m and 50 m depth). When chlorophyll is considered as a proxy of NPP, vertically integrated chlorophyll is more relevant. The vertical-averaged (5 m to 150 m) chlorophyll in the model and observation are 0.1595 and 0.1668 mg m$^{-3}$, respectively, which have a marginal difference (< 5%). Both the modeled and observed chlorophyll concentrations have a large range from 0 to >1.0 mg m$^{-3}$ in the subsurface chlorophyll maxima. This reflects the large spatial variability in chlorophyll.

Following the analysis in Sect. 3.1, the multi-variable regression analysis on the model outputs were also conducted. The modeled NPP presents a phase lag with respect to the UI and KE variation. When NPP is lagged for one month, the correlation is 0.752 with a *p*-value of 0.0214, suggesting a significant regulation of the physical forcing to the productivity. Additionally, the composites of the HNA, LNA, and normal scenarios (Fig. 6c) based on model outputs

show contrasts among scenarios comparable to those in the observed cases in Fig. 4, further suggesting the reasonability of the model simulation (Fig. 8).

In summary, one could find that our model performs reasonably in reproducing the key spatiotemporal features in the hydrodynamics and ecosystem of VBUS. Inevitably, some discrepancies exist, which are less evident in the summer months. Nevertheless, considering the focus is to investigate the positive correlation between the productivity and the circulation, which was captured by the model (Fig. 8), these shortcomings could be accepted.

**3.3 Analysis of Model Results**

Modeled circulation and potential density from the multi-summer average are presented in Fig. 9a-d, with sea surface height overlaid. In the upwelling region, the doming of isopycnals (Fig. S2) is discernable as in the classical coastal upwelling models (O'Brien and Hurlburt, 1972). Consistent with previous studies, the coastal current flows northward along the shelf (Hein et al., 2013). The current also disperses freshwater from the Mekong River, while the water seldom spreads away from the coast. The coastal current veers at ~11° N, directs offshore and then separates, forming the quasi-stationary anticyclone centered at ~110° E, 9° N. Near the core of the anticyclone, vigorous vertical motion near the surface can be found, implying submesoscale processes in play. Near 108° E, intensive onshore flow ascends on the slope. The high-density bottom water outcrops at 107° E, rejoining the coastal water and directing north, thus forming a circuit.

The biogeochemical variables reveal that the ecosystem is largely controlled by the circulation (Fig. 9e-h, Fig. S2). Lateral nutrient gradient appears at the periphery of the anticyclone, which is characterized with depressed nitrate isosurface in the core and domed isosurface due to the upwelling and river injection near the coast (Fig. 9e). Stimulated by the river-injected and locally upwelled nutrient near the coast, primary production (PP) shows a surface maximum of >30 mg C m$^{-3}$ d$^{-1}$ (Fig. 9g). The water with high production is then advected offshore by the jet (Fig. 9h), leading to an offshore bloom patch in a curved shape which is familiar in the Vietnam coast (e.g., Fig. 5c). The jet also conveys the water with high particulate organic carbon offshore. The distribution of particulate organic carbon is somewhat deeper and more spread than that of high PP water, suggesting the vertical sinking and lateral transport processes (Nagai et al., 2015). Remineralization of organic carbon results in a subsurface ammonium maximum at ~50 m (Fig. 9f) consistent with other studies in SCS (Li et al., 2015). Part of the ammonium could then fuel nitrification and production, while the rest rejoins the circulation with the upwelling water in the bottom Ekman layer. In summary, the model outputs clearly reveal circuiting circulation and cycled ecosystem, which will be further discussed in Sect. 4.

## 4    Discussion

### 4.1 Biogeochemical Cycle in VBUS

In the summer VBUS system, it is generally agreed that the wind's predominant role in controlling the variability in the production of VBUS, especially on the inter-annual scale (Dippner et al., 2013). This is also the case in our analysis where UI contributes ~45% of the total variability in production. In addition, via analysis of satellite data and model outputs, consistent and robust positive contribution from the local circulation to the biological production was also revealed. The contribution of the circulation is distinct from the major coastal upwelling systems, where the offshore

transport by the mean current appears to suppress the production by reducing the nearshore nutrient inventory (Gruber et al., 2011; Nagai et al., 2015). The separated current system was considered to transport high-chlorophyll water offshore (Xie et al., 2003). In the offshore region, the production appeared to be elevated (Bombar et al., 2010). However, the fate of the offshore nutrients was rarely investigated in the literature.

Comparing the ecosystems in LNA and HNA (Fig. 10), the following stages of the cycle can be deduced: (1) The

upwelled and riverine input nutrient (majorly inorganic) stimulate high production near the Vietnam coast (Dippner et al., 2007); (2) The produced organic matters are transported offshore by the jet; The water has high chlorophyll (e.g., Fig. 2a) and high organic matters (Fig. 9h) in the euphotic zone; (3) A significant portion of the nutrient (majorly in organic form) is transported back to the south of VBUS by the westward recirculation. The quasi-stationary rotating anticyclone impedes further offshore leakage of the nutrients (Fig. 10i). (4) The trapped organic matters are

remineralized, forming the subsurface maxima of ammonium (Fig. 9f, which supports regeneration production with an $f$-ratio of ~60%, Table 1) and replenishing the nitrate by nitrification. The offshore remineralization can be supported by the high oxygen consumption found off Vietnamese coasts (Jiao et al., 2014). Afterward, the nutrients are upwelled by bottom Ekman pumping (see high nutrient in the bottom boundary layer in Fig. 9e) and wind-induced upwelling, and finally rejoin in the local biogeochemical cycle. The recirculation may determine the available nutrient

inventory, therefore playing a significant role in controlling the productivity, which will be further supported by the experiment in the following section.

### 4.2 Dynamic Analysis

By controlling the available nutrients, the recirculation modulates the productivity in VBUS. The influence of the recirculation is further elucidated below. Table 1 summarizes the difference of the ecosystems in the standard run and

NO_ADV experiment. The NO_ADV experiment can be regarded as an extreme case where the circulation shows very weak tendency of separation without the recirculation (also see Fig. S1), which is representative for the LNA scenario. In terms of force balance, the recirculation is maintained by the balance between the inertial force and the Coriolis force. Without the nonlinear term, this balance could not exist. The horizontal and vertical fluxes of nitrate in three scenarios are also depicted in Fig. 11.

315       In the VBUS, similar to other coastal upwelling systems (Nielsen and Jensen, 1957), the availability of nutrients principally controls the productivity. Considering a quasi-steady state of nutrient in a coastal region, river-injected and upward inputted nutrient should be counter-balanced by vertical export production and lateral exchanges. The lateral exchanges include both advection and diffusion, while it was pointed out that horizontal mixing is one or two order-of-magnitude lower than that of horizontal advection (Lu et al., 2015). Hence, as a sink term, the lateral

exchanges are determined mostly by the advective fluxes normal to the boundary of the predefined box. Diagnostic also suggests a dominance role of advection process in the vertical over mixing. Given the fact the standard run and NO_ADV experiment have the same riverine input and similar export flux, one could infer that the difference between two model cases is largely due to the different lateral transports and upwelled fluxes of nutrients.

      In the LNA (Fig. 4a and Fig. 8a) and also in the NO_ADV experiment (Fig. S1), the circulation pattern switches

to the along-isobath pattern, which modifies the local biogeochemical cycle. More nutrient is transported northward and offshore out of VBUS and never comes back, leading to a reduction of nutrient (Fig. 11). This effect can be demonstrated by cross-section nutrient flux across 109° E section. The more nutrient leakage, the less westward nutrient flux across this section. In NO_ADV run, the westward flux of nitrate is significantly reduced by 36.2% (Table 1). The reduction of nutrient is accompanied with suppressed the upward nutrient flux (-46.5%) near the shelf

edge (~100 m). As a consequence of more leakage and less upwelling influx, the nitrate reservoir and new production are significantly reduced by 20.7 % and 21.9 %, significantly inhibiting the primary production process (15.7 %, Table 1). Other ecosystem constituents decrease to a limit degree, such as -2.6% for ammonium, and -3.0% for DOC. This interpretation is further supported by the post-El Niño scenario. In 2010, more significant suppression occurs in the vertical nutrient flux (-99.6%), while the horizontal fluxes also respond to decrease. Due to the drawdown in the wind-

induced upwelling and recirculation (Table 1), the production is extremely low in summer 2010 (Fig. 3c).

      The larger KE, the more intensive separation (see Appendix B for additional discussion about this point). This separation was similar to the Gulf Stream detachment problem in many aspects. In particular, various factors could affect the intensity and position of the separation (Chassignet and Marshall, 2008). Generally, the mechanism of the current separation was considered to be ascribed to the wind stress curl (Xie et al., 2003). We also found that the

current separation was very sensitive to the wind stress curl (not shown in figures). Hence, one may argue that this co-variability is controlled by the variation of wind-forcing. However, as mentioned in Sect. 3.1, a large part (~68%) of the variation in KE is unexplained by the large-scale wind. In addition, our sensitivity experiments without the nonlinear advection of momentum showed very weak separation, in agreement with that of Wang et al. (2006). This implies wind-forcing is not the only factor in controlling the current system, while the intrinsic dynamics are also

important. The accelerated coastal current is also associated with intensified cross-isobath transport by bottom Ekman effect (Gan et al., 2009) or dynamic upwelling (Yoshida and Mao, 1957) due to the rotational current (Dippner et al., 2007). Hence, low KE reduces the recirculation of nutrients. Combining all the effects, the intensified current is a condition favorable for the nutrient inventory, and hence the productivity. To a larger scale, the recirculation current couples coastal upwelling and offshore region in the major coastal upwelling systems, e.g., in the Canary basin (Pelegrí

et al., 2005). In this study, this effect was also found to be important, which may contribute up to 15% of the productivity variability.

## 5    Conclusions

Via analyzing the summertime remote sensing data in the VBUS, a tight spatiotemporal covariation between the ecosystem and near-surface circulation was revealed. The water with high kinetic energy appeared to coincide with
high chlorophyll variability. Statistical analysis suggested that the high level of productivity was associated with the high level of circulation intensity, which accounted for ~15% of the variability in productivity. Elevated kinetic energy and intensified circulation were related with the separation of the upwelling current system. Especially, in the low-productivity scenarios, the circulation pattern shifts from the intensive separation pattern to a moderate alongshore non-separated pattern.

A physical-biological coupled model was applied to investigate the positive contribution from the circulation intensity to the productivity. A numerical experiment was also designed to reproduce the weak-separated circulation pattern without the recirculation. Inspection into the model results highlighted the recirculation's role in the local biogeochemical cycle. As presented in the schematic diagram in Fig. 12, the separated circulation and resultant recirculation were favorable for the nutrient inventory. During non-separation scenarios, the nutrients northward
transported by the alongshore current would never come back, leading to a nutrient leakage. The nutrient leakage further induced the feedback summarized in Fig. 12b, which could reduce the nitrate inventory by ~21% and the NPP by ~16% in the experiment representative for very weak separation. The weakened coastal current was also associated with reduced dynamic upwelling, hence further reducing the vertical flux of nutrient. This resulted in the positive contribution to the productivity.

This finding provides a new insight into the complex physical-biological coupling in the Vietnam coastal upwelling system. Moreover, this understanding could help to predict the future reaction of productivity in the SCS. As revealed by Yang and Wu (2012), the summertime near-surface circulation of SCS had experienced a long-term trend of being more energetic, characterized with intensified separation and recirculation in the VBUS (see their figure 9). Whether this long-term trend of circulation will also induce a potential trend in the ecosystem in response to future climate
changes is a topic of common interests, which merits further investigation.

## 6    Data Availability

The CCMP gridded Ocean Surface Wind Vector L3.0 First-Look Analyses (Version 1) data was accessed [2015-03-12] at http://dx.doi.org/10.5067/CCF30-01XXX. The MODIS Aqua Level 3 chlorophyll data was accessed [2014-05-16]    at    http://oceancolor.gsfc.nasa.gov.    The    VGPM    NPP    data    was    available    at
http://www.science.oregonstate.edu/ocean.productivity/index.php. Gridded monthly-mean ADT, available at http://www.aviso.altimetry.fr/en/data/products/sea-surface-height-products/global/, was produced by Ssalto/Duacs (http://www.aviso.oceanobs.com/duacs/), and was distributed by Aviso with support from the Centre National d'Etudes Spatiales (*Cnes*). The OISST data was obtained from the National Climatic Data Center of NOAA

(https://www.ncdc.noaa.gov/oisst/data-access). Atmospheric forcing data was acquired from the National Centers for

Environmental Prediction Reanalysis data (http://www.esrl.noaa.gov/psd/data/gridded/data.ncep.reanalysis.html) distributed by the NOAA/OAR/ESRL PSD, Boulder, Colorado, USA (http://www.esrl.noaa.gov/psd/). The WOA data distributed by National Climatic Data Center of NOAA was available at https://www.nodc.noaa.gov/OC5/woa13/. The river runoff data was available at http://www.cgd.ucar.edu/cas/catalog/surface/dai-runoff/.

**Appendix A: Abbreviations**

In Table A1, all the acronyms in the paper were listed.

**Table A1 List of Acronyms**

| Acronym | Definition | Acronym | Definition |
| --- | --- | --- | --- |
| SCS | South China Sea | NPP | vertical-integrated Net Primary Production |
| VBUS | Vietnam Boundary Upwelling System | PP | Primary Production (as a function of depth) |
| CCMP | Cross-Calibrated Multi-Platform data | KE | kinetic energy |
| MODIS | Moderate Resolution Imaging Spectroradiometer data | TFOR | Taiwan Strait Nowcast\Forecast system |
| VGPM | chlorophyll-based Vertically Generalized Production Model | CoSINE | Carbon, Silicon, Nitrogen Ecosystem model |
| ADT | Absolute Dynamic Topography | NO_ADV | model experiment with no advection term in momentum equations |
| OISST | Optimum Interpolation Sea Surface Temperature | UI | Upwelling Intensity |
| HNA/LNA | high/low-NPP anomaly scenario | | |

## Appendix B: Flow Separation and Kinetic Energy

Qualitatively, both the analysis based on remote sensing data and model results suggest the separation flow is linked with stronger KE (~65% larger in HNA case then LNA case, Sect. 3.1). Moreover, a separation index (SI) is defined to quantitatively explain the relation between the flow separation and intensified circulation. The SI can be written as:

$$SI = \sum \frac{u \cdot \cos\varphi + v \cdot \sin\varphi}{\sqrt{u^2 + v^2}}, \tag{A1}$$

Where $u$ and $v$ are the two surface velocity components, and $\varphi$ is the angle between the topography gradient and the positive $x$-axis. This SI is essentially the area-averaged cross-isobath velocity normalized by the magnitude of the velocity, which is used to quantify flow separation here.

Fig. A1 shows the spatial distribution of SI in Aug 2010. The positive values indicate the flow is downslope while negatives suggest ascent. Large SI can be observed near the separation point ~11.5°N. Taking spatial average over the box region in Fig. A1, a generally good positive correlation (R=0.7175, p<0.01) between log(KE) and SI is found (see Fig. A2). The log(KE) and SI presents a logistic-type relationship, where SI asymptotically approaches a maximum value of ~0.35. This suggests that the strong flow separation and elevated KE are tightly linked. Further, from the scatter plot of KE vs. SI (Fig. A2), we find that 0.1 $m^2s^{-2}$ is a critical value, which divides the data into two subsets while minimizes the slope of the right part (blue fitting curve in Fig. A2).

Fig. A1 shows the distribution of SI in Aug 2010. Positive values indicate that the flow is separating and downslope, and that may be seen off Vietnam south of the coastline bend. Large SI (~1.0) can be observed near the separation point ~11.5°N. Taking spatial average over the box region in Fig. 2a or Fig. A1, there is a good positive correlation (R=0.7175, p<0.01) between log(KE) and SI (see Fig. A2). Moreover, SI may be seen to generally increase with KE to a value of 0.25~0.3 and then it levels off (i.e. the slope becomes less) – see the red and blue lines in Fig. A2. The log(KE) and SI thus appear to show a logistic-type behavior, in which SI asymptotically approaches some maximum value (in this case ~0.3). This suggests that the strong flow separation and elevated KE are tightly linked. From Fig. A2, the value of KE $\approx$ 0.1 $m^2s^{-2}$ appears to be a critical value.

Dynamically, the nonlinear advection term in the momentum equation can be written as the vector invariant form [see e.g. (Gill, 1982)]:

$$\vec{u} \cdot \nabla\vec{u} = (\nabla \times \vec{u}) \times \vec{u} + \nabla(\frac{1}{2}|\vec{u}|^2), \tag{A2}$$

This decomposition directly links the nonlinear advection term and the gradient of KE (which scales KE over a length scale L). Meanwhile, the nonlinear advection is an important mechanism in driving flow separation [see, for instance, Oey et al. (2014)]. Stronger advection suggests intense cross isobath flow. Therefore, a dynamic linkage between the flow separation and the intensified KE and circulation can also be established, further supporting this argument.

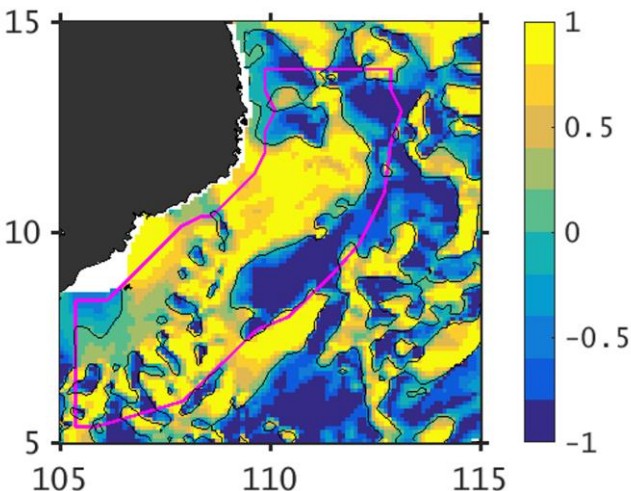

Figure A1 Example of modeled SI in Aug 2010.

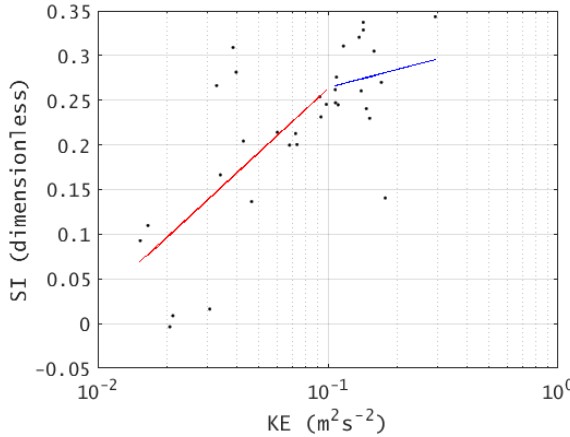

Figure A2 Summer-month (MJJAS) KE vs SI averaged over the box region in Fig. A1 (overall R=0.7175, p<0.01).

**Supplementary**

**Table S1 Initial values for some of the ecosystem variables**

| Name | Description | Initial value | Unit | Note |
|------|-------------|---------------|------|------|
| S1_N | Nitrogen for pico-phytoplankton | 0.04 | mmol N m$^{-3}$ | |
| S1_C | Carbon for pico-phytoplankton | 0.265 | mmol C m$^{-3}$ | S1_N/16*106 |
| S1CH | Chlorophyll for pico-phytoplankton | 0.06 | mg m$^{-3}$ | |
| S2_N | Nitrogen for diatom | 0.08 | mmol N m$^{-3}$ | |
| S2_C | Carbon for diatom | 0.53 | mmol C m$^{-3}$ | S2_N/16*106 |
| S2CH | Chlorophyll for diatom | 0.12 | mg m$^{-3}$ | |
| S3_N | Nitrogen for coccolithophorids | 0.04 | mmol N m$^{-3}$ | |
| S3_C | Carbon for coccolithophorids | 0.265 | mmol C m$^{-3}$ | S3_N/16*106 |
| S3CH | Chlorophyll for coccolithophorids | 0.06 | mg m$^{-3}$ | |
| Z1_N | Nitrogen for small zooplankton | 0.02 | mmol N m$^{-3}$ | |
| Z1_C | Carbon for small zooplankton | 0.1325 | mmol C m$^{-3}$ | Z1_N/16*106 |
| Z2_N | Nitrogen for meso-zooplankton | 0.02 | mmol N m$^{-3}$ | |
| Z2_C | Carbon for meso-zooplankton | 0.1325 | mmol C m$^{-3}$ | Z2_N/16*106 |
| DD_N | Detritus nitrogen | 0.02 | mmol m$^{-3}$ | |
| DD_C | Detritus carbon | 0.1325 | mmol m$^{-3}$ | DD_N/16*106 |
| BAC_ | Bacteria carbon | 0.01 | mmol C m$^{-3}$ | |
| DDCA | Detritus inorganic carbon | 0.01 | mmol m$^{-3}$ | |
| DDSi | Detritus silicate | 0.03 | mmol m$^{-3}$ | |


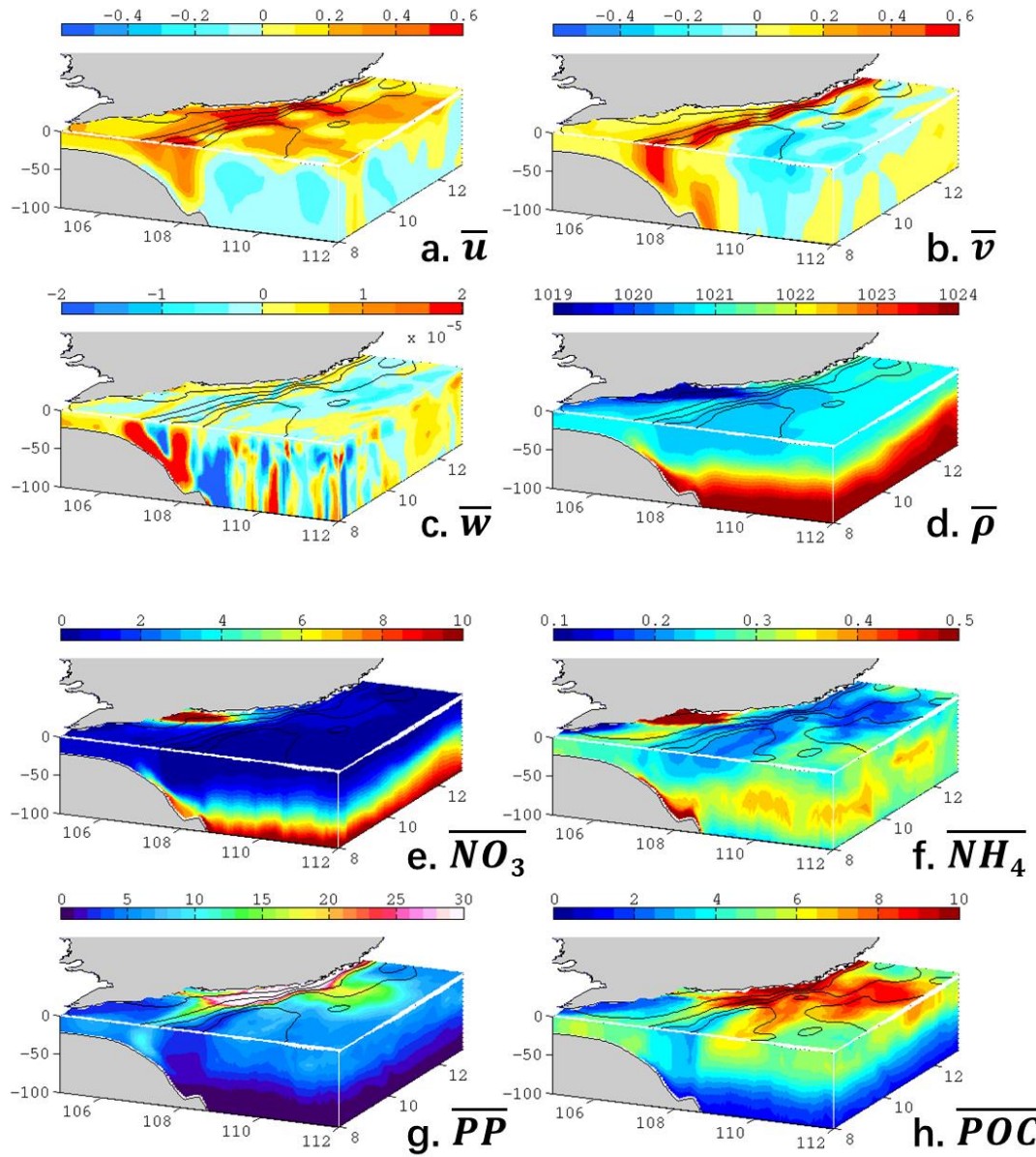

Figure S1 Same with Fig. 9, but for NO_ADV model run. (a) Zonal velocity in m s⁻¹, (b) meridional velocity in m s⁻¹, (c) vertical velocity in m s⁻¹, (d) potential density in kg m⁻³, (e) nitrate in mmol m⁻³, (f) ammonium in mmol m⁻³, (g) primary production in mg C m⁻³ d⁻¹, and particulate organic carbon in mmol C m⁻³. Overlapped contours are the mean sea level (every 0.1 m).


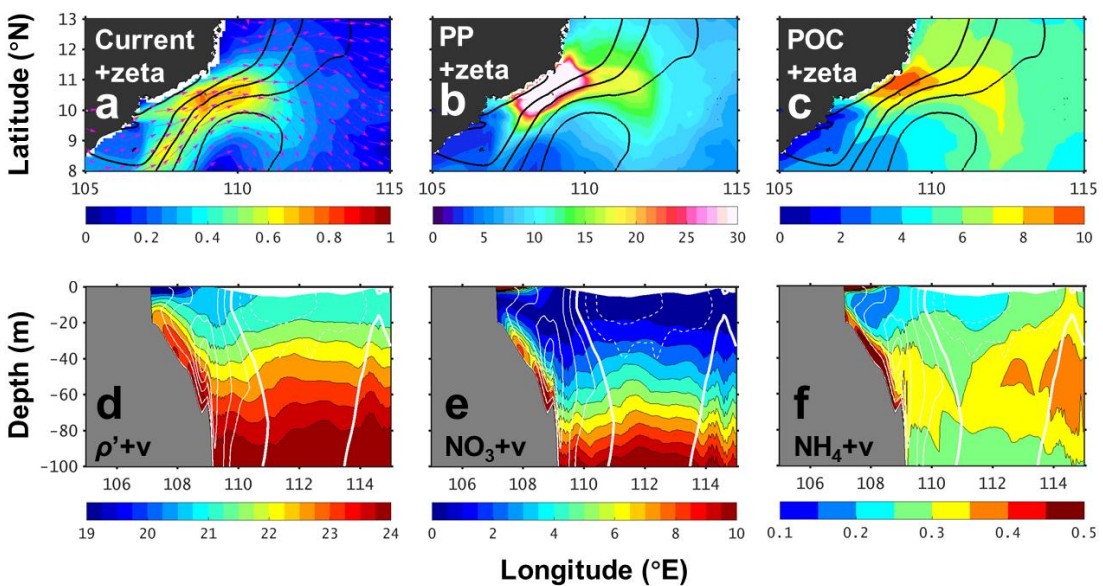

Figure S2 (a-c) Modeled sea level (black contour, CI=0.1 m) overlaid with (a) surface current (color: magnitude in
m s$^{-1}$; vector: flow direction), (b) surface primary production (mg C m$^{-3}$ d$^{-1}$), and (c) particulate organic carbon
(mmol C m$^{-3}$). (d-f) Sections along 10° N: meridional velocity (positive in solid contours and negative in dashed,
CI=0.1 m s$^{-1}$. Thick contours indicate zero value) overlaid with (d) potential density anomaly, (e) nitrate
concentration (mmol m$^{-3}$), and (f) ammonia concentration (mmol m$^{-3}$).

**Acknowledgements**

This study was supported by grant No.2016YFA0601201 from the Ministry of Science and Technology of the People's Republic of China (MOST), and grants 41476005, 41476007, 41876004 and 41630963 from the National Natural Science Foundation of China (NSFC). The authors would like to thank Dr. Fei Chai for providing the code of CoSINE model. The authors would also like to thank the three reviewers for their useful comments.

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

**Table**

**Table 1 Summery of the ecosystems in three model scenarios**

| Quantities integrated over top 100 m of the box region (Fig. 2b) | Normal years (except 2010) | NO_ADV | 2010 post-El Niño |
|---|---|---|---|
| $NO_3$ ($\times 10^9$ mol) | 11.3 | 8.96 (-20.7%) | 10.71 (-5.2%) |
| $NH_4$ ($\times 10^9$ mol) | 1.52 | 1.48 (-2.6%) | 1.51 (-0.7%) |
| DOC ($\times 10^9$ mol C) | 234 | 227 (-3.0%) | 236 (+0.9%) |
| Particulate Organic Carbon ($\times 10^9$ mol C) | 22.7 | 22.4 (-1.3%) | 19.9 (-12.3%) |
| NPP (mmol N m$^{-2}$ d$^{-1}$) | 4.65 | 3.92 (-15.7%) | 3.57 (-23.2%) |
| New Production + Regeneration Production (mmol N m$^{-2}$ d$^{-1}$) | 2.83+1.82 | 2.21+1.71 (-21.9%, -6.0%) | 1.82+1.75 (-35.7%, -3.8%) |
| **Fluxes** | | | |
| Vertical $NO_3$ flux across 100 m level ($\times 10^9$ mol d$^{-1}$, positive upward) | 0.2454 | 0.1313 (-46.5%) | 0.0011 (-99.6%) |
| Top 100 m integrated zonal $NO_3$ flux across 109° E section[*] ($\times 10^9$ mol d$^{-1}$, positive westward) | 0.4156 | 0.2652 (-36.2%) | 0.2013 (-51.6%) |
| Vertical volume flux across 100 m level (Sv, positive upward) | 0.22 | 0.14 (-36.4%) | 0.04 (-81.8%) |
| Top 100 m integrated zonal volume flux across 109° E section[*] (Sv, positive westward) | 0.44 | 0.01 (-97.7%) | 0.28 (-36.4%) |

* See Fig. 11a for the location.


**Figures**

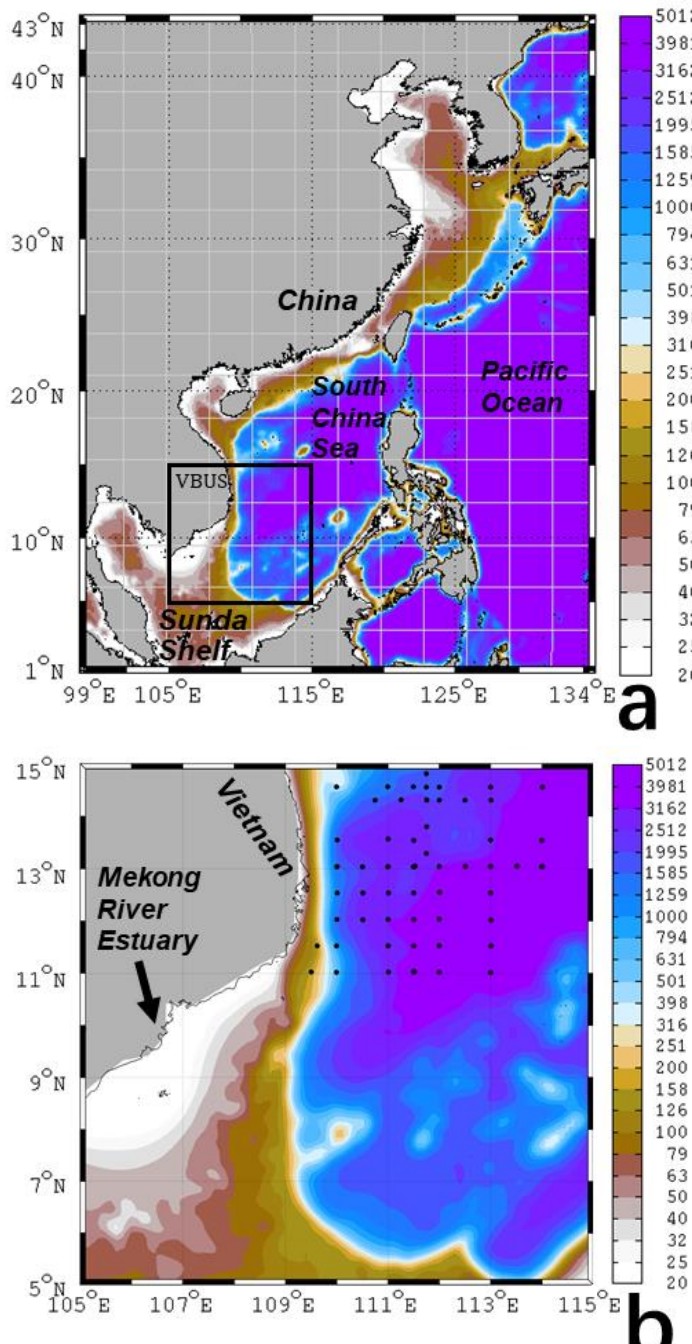

**Figure 1 (a) Model domain and the bathymetry (unit: meter) for the TFOR-CoSINE model. Model grid nodes are shown every 25 points. The study area VBUS is boxed. (b) Zoom-in area of VBUS. Black diamonds are the observation stations (see text).**

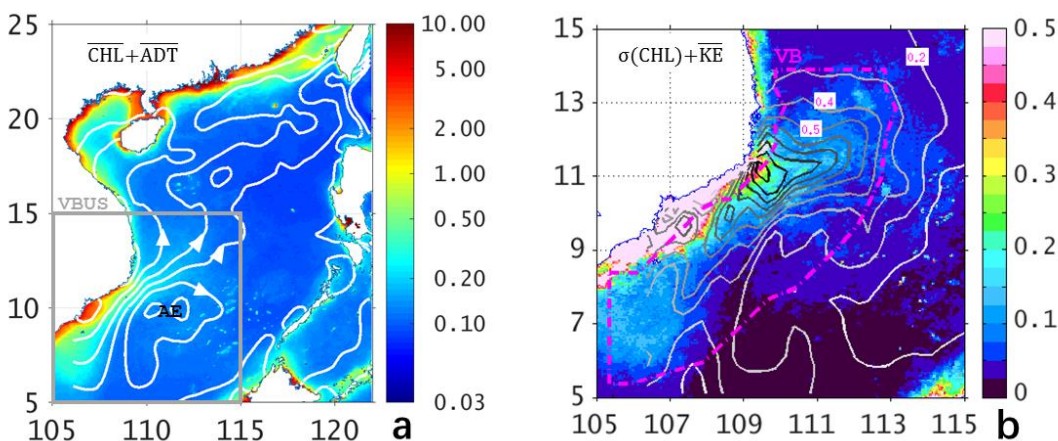


**Figure 2 (a) Summertime (MJJAS) average of surface chlorophyll concentration (color shading, unit: mg m⁻³) from MODIS, overlapped white contours are mean ADT with the arrows showing the directions of surface currents. Gray box is the region of interest (VBUS), while AE shows the center of the anticyclone. (b) Standard derivation of surface chlorophyll (color shading, unit: mg m⁻³) overlapped with the contours of surface KE with an interval of 0.1 from 0.1 to 1.0 (unit: m² s⁻²). The magenta dot-dash contour delimits the ocean region over which chlorophyll and KE are averaged (see text).**


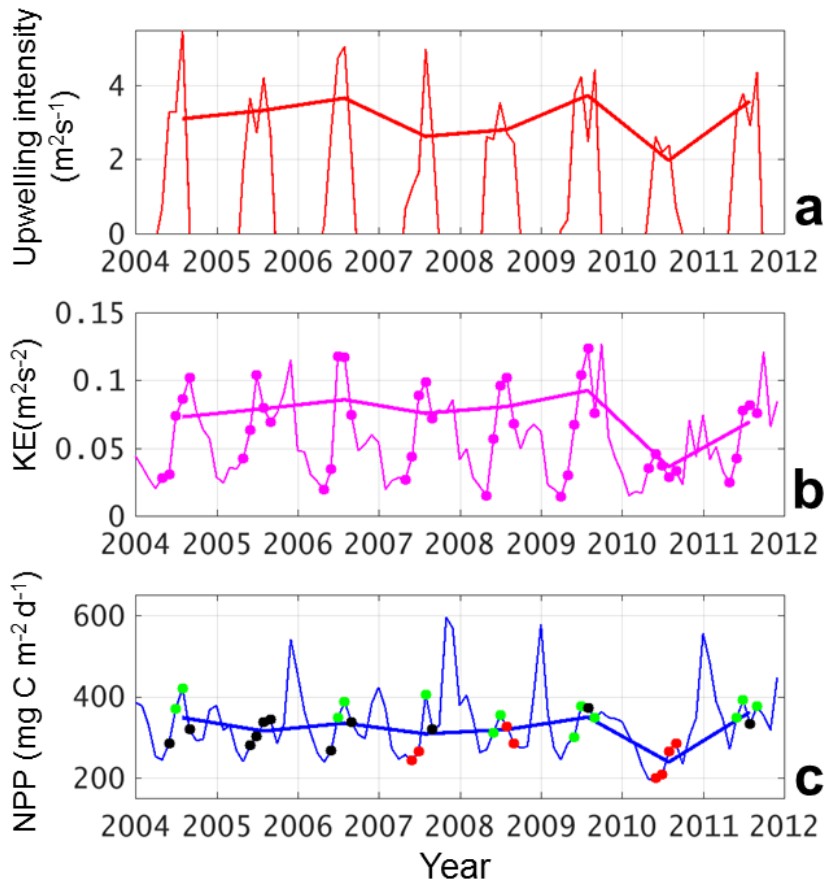

**Figure 3** Time series of (a) UI in $m^2 s^{-1}$, (b) KE in $m^2 s^{-2}$ and (c) NPP in mg C $m^{-2} d^{-1}$ of monthly data (thin lines) and summer mean (thick lines). In (a), only the positive (upwelling-favorable) values are shown. In (b), months with positive UI are marked with dots. In (c), green, black and red dots indicate HNA, normal and LNA scenarios (see text for definition).


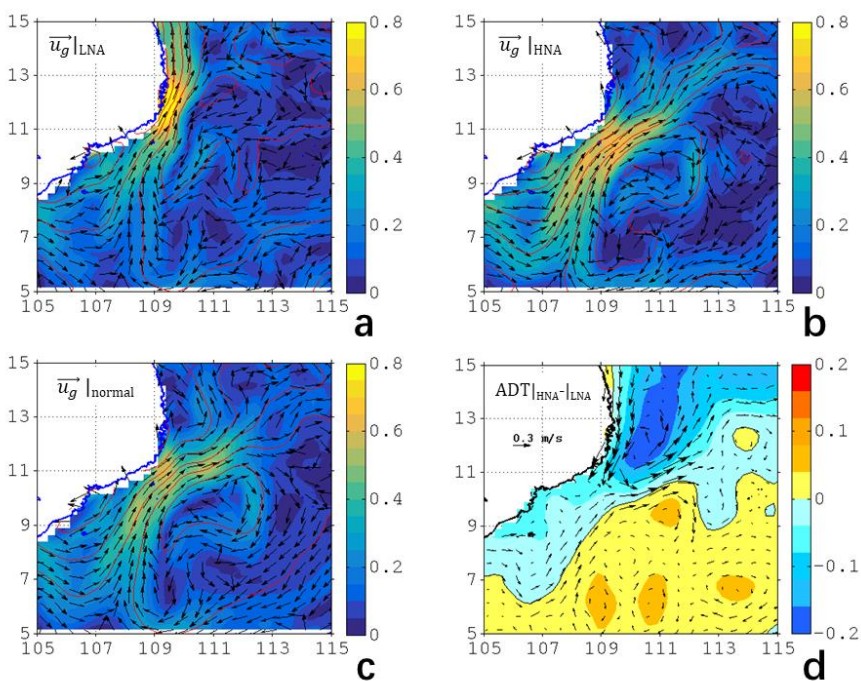

**Figure 4 The surface current velocity (color shading, unit: m s⁻¹), direction (vectors), and respective ADT (red contours, unit: meter) in (a) LNA, (b) HNA, and (c) normal months (i.e., neither LNA nor HNA) scenarios (see text for criteria). (d) The differences of ADT and current between HNA and LNA.**


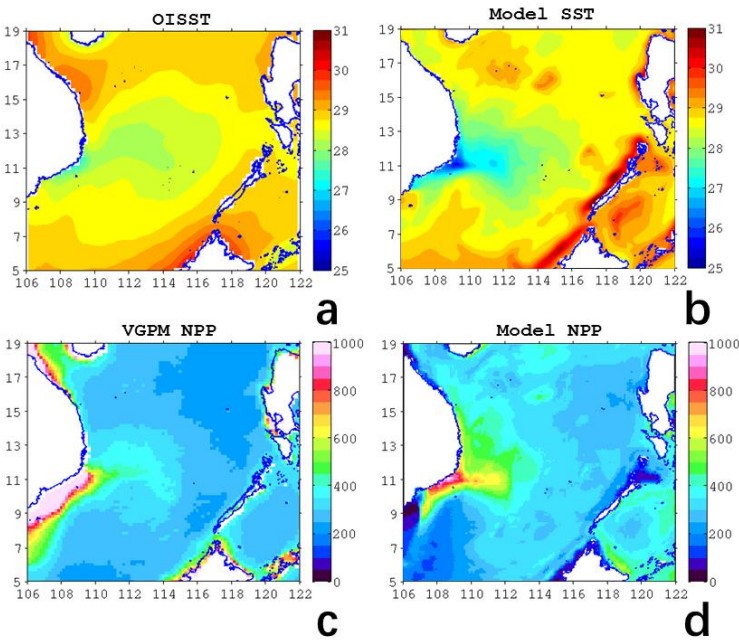

**Figure 5 (a) OISST and (b) model SST (unit: °C), (c) VGPM NPP and (d) modeled NPP (unit: mg C m$^{-2}$ d$^{-1}$) in multi-year August average.**

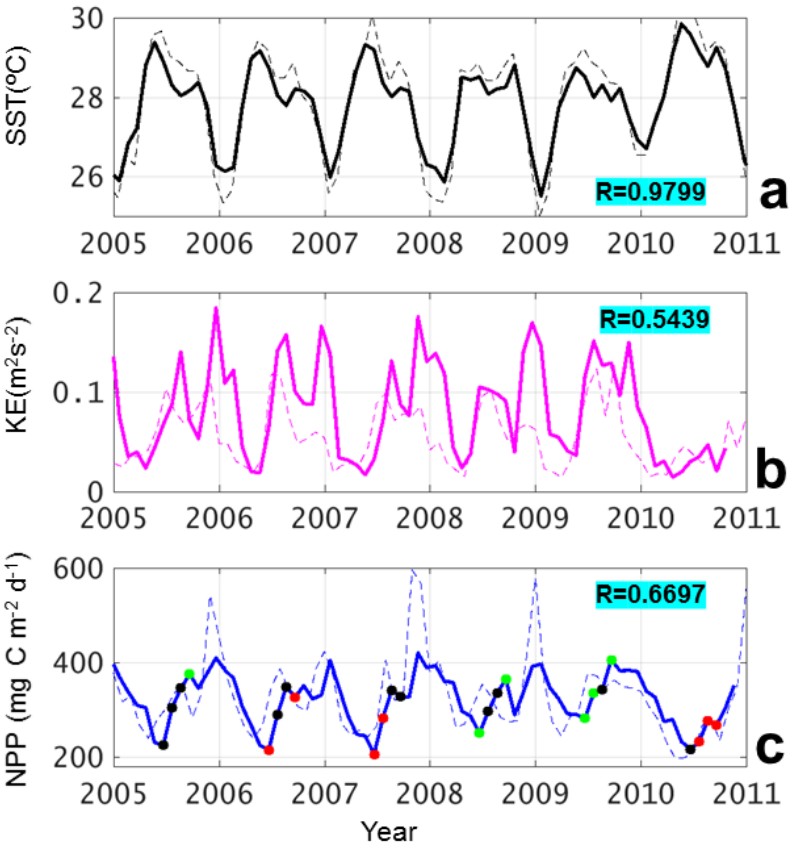


**Figure 6 Thick lines: modeled (a) SST in °C, (b) KE in m² s⁻², and (c) NPP in mg C m⁻² d⁻¹ averaged over the box region (see Fig. 2b), with respective observation data (thin dashed lines). Correlation coefficients are also show in each plot. In (c), green, black and red dots indicate HNA, normal and LNA scenarios (see text for definition).**


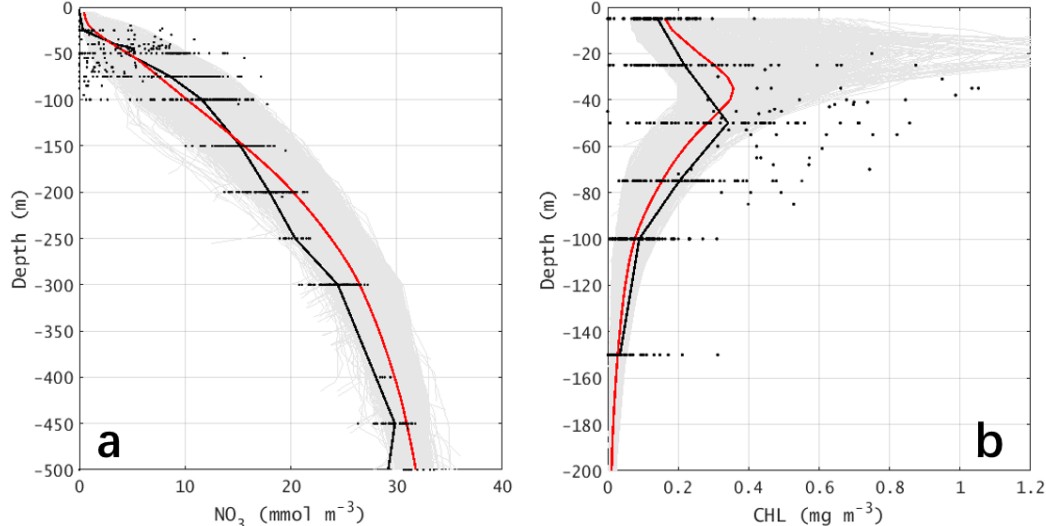

**Figure 7 The vertical profiles of (a) nitrate concentration (unit: mmol m⁻³) and (b) chlorophyll concentration (unit:mg m⁻³). In both plots, the black dots are the observation values (see Fig. 1b for stations). The gray area are the envelop for all model stations in the same area and month, while the red lines are the area-mean profiles.**


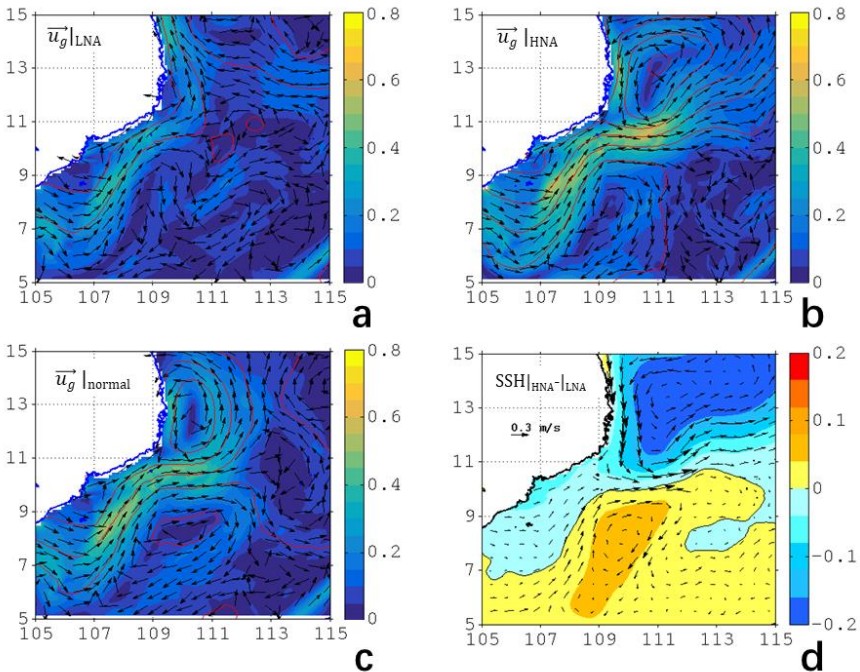

**Figure 8 Same with Fig. 4, but based on model outputs.**

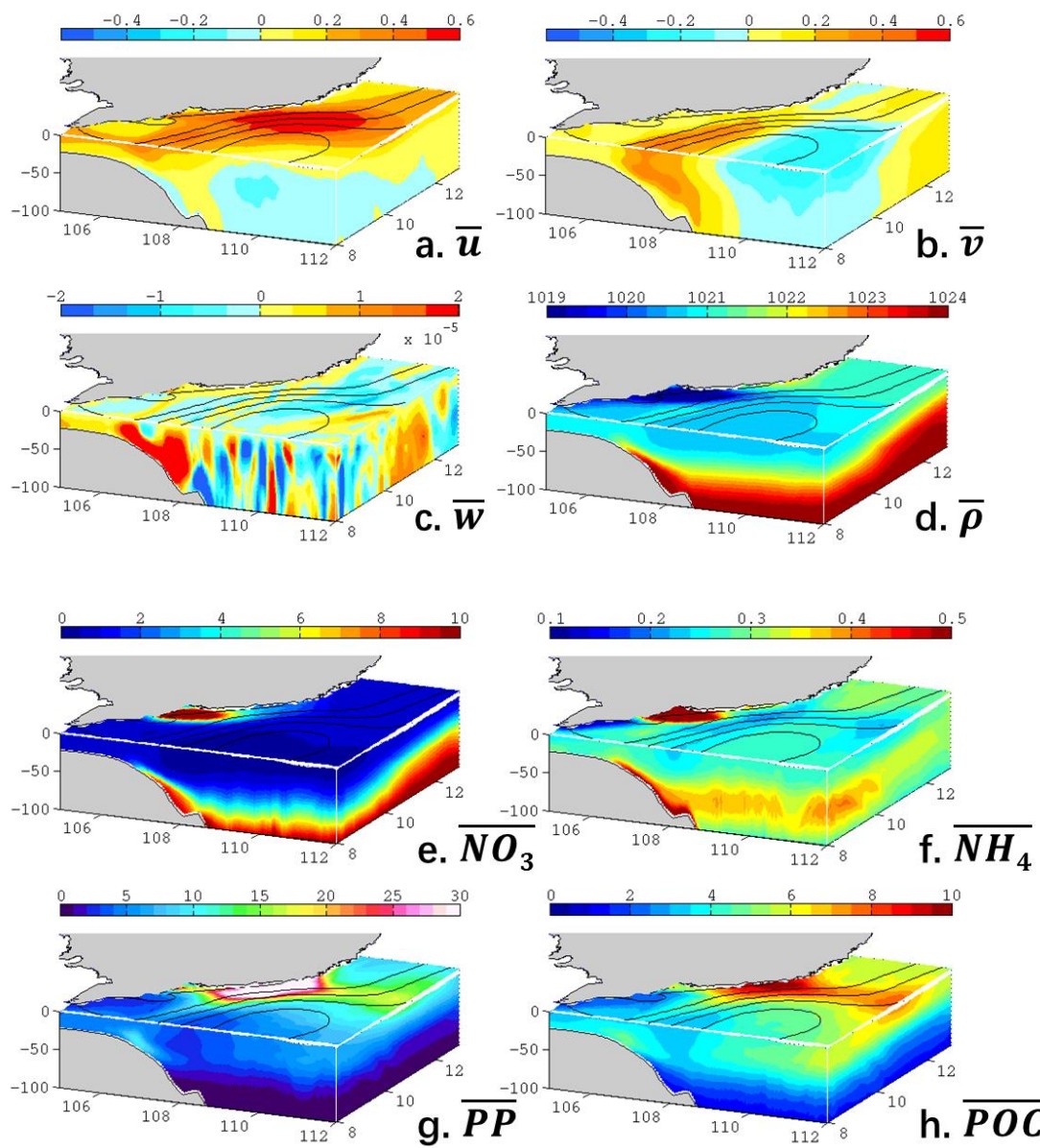


**Figure 9 Three-dimensional distribution (standard run) of the summer mean (a) zonal velocity in m s⁻¹, (b) meridional velocity in m s⁻¹, (c) vertical velocity in m s⁻¹, (d) potential density in kg m⁻³, (e) nitrate in mmol m⁻³, (f) ammonium in mmol m⁻³, (g) primary production in mg C m⁻³ d⁻¹, and particulate organic carbon in mmol C m⁻³. Overlapped contours are the mean sea level (every 0.1 m).**


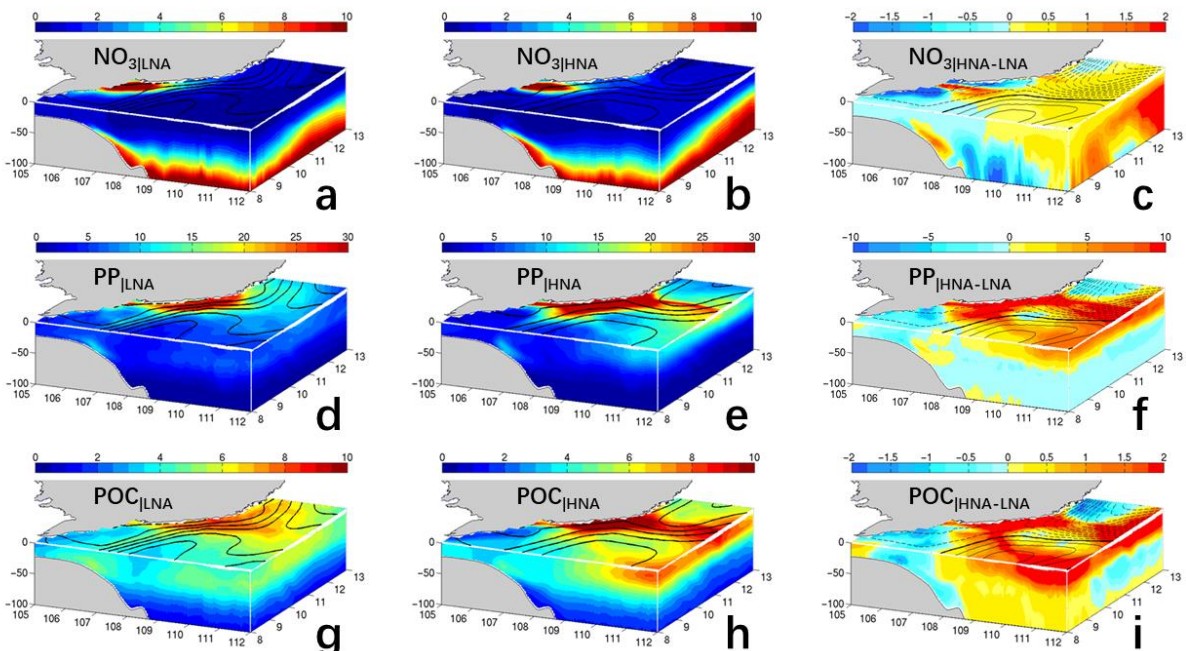

**Figure 10 Modeled NO₃ (first row, in mmol m⁻³), PP (second row, mg C m⁻³ d⁻¹) and particulate organic carbon (third row, mmol C m⁻³) distribution in LNA (left column), HNA (middle column) and the difference between two scenarios (right column). See text for the defination of LNA and HNA.**


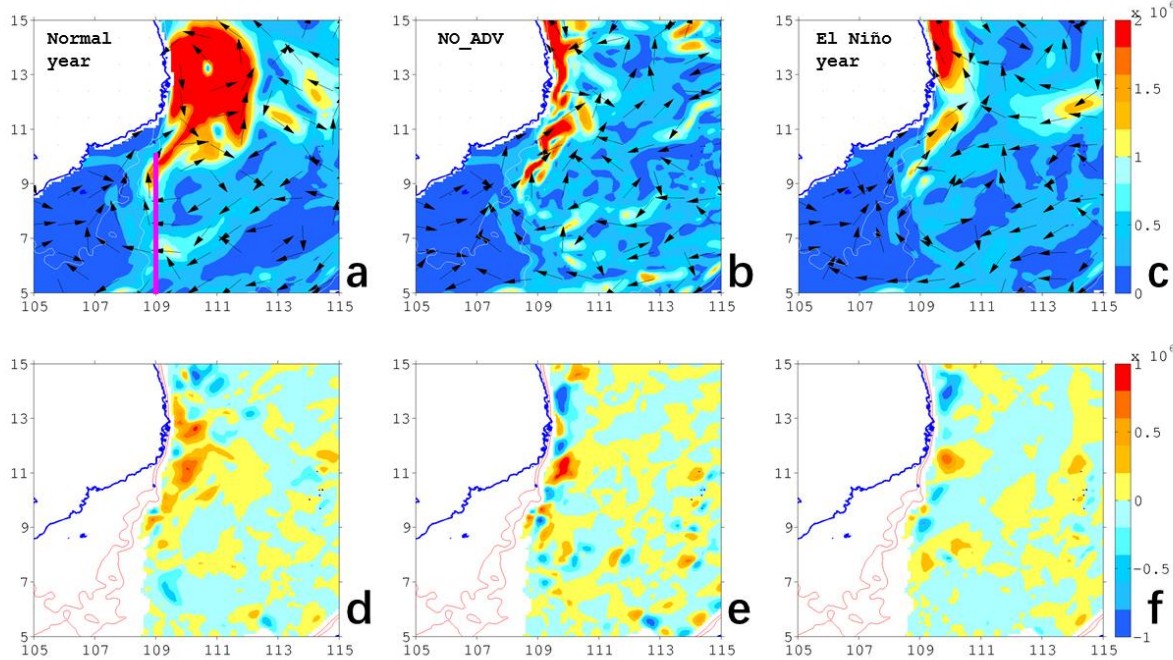

**Figure 11 (Upper) Modeled 0-100 m integrated nitrate fluxes (unit: mmol s⁻¹) in horizontal plane. Color shading is the magnitude while vectors denote the direction. (Lower) Vertical flux across 100 m level for normal year (a and d, years other than 2010), NO_ADV case (b and e), and post-El Niño (c and f, in year 2010). Overlapped contours are the 50 m and 75 m isobath. In (a), the magenta line is the 109° E section (Table 1).**


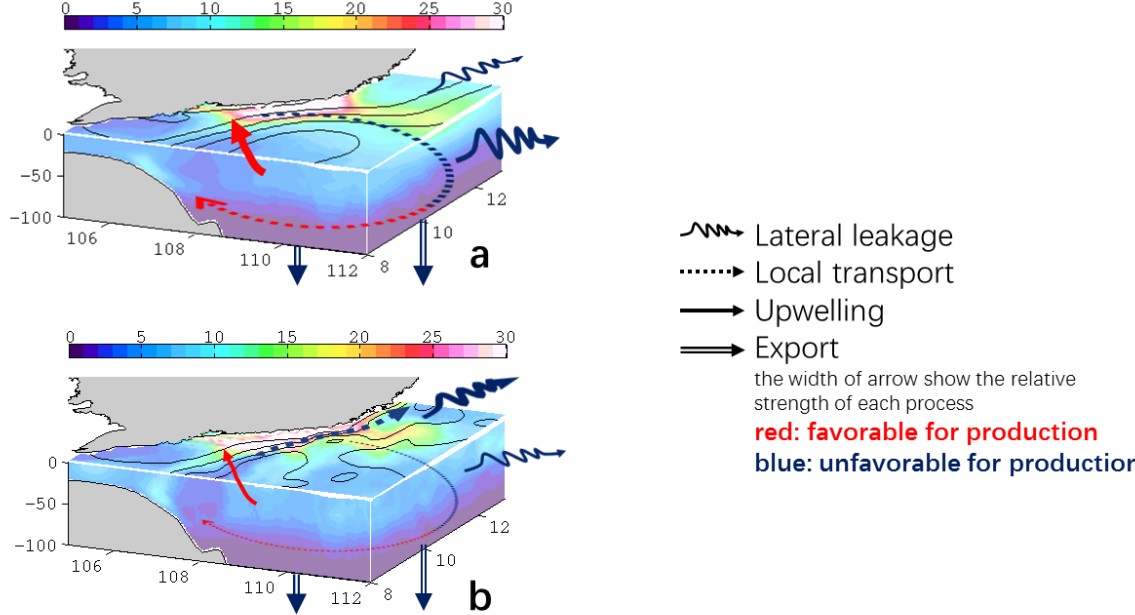

**Figure 12 Schematic diagram summarizing the dynamics in different scenarios of distinct circulation pattern in the VBUS, overlapped with the three-dimensional distribution of PP (unit: mg C m$^{-3}$ d$^{-1}$). (a) Normal state: The separated jet transports**
**the upwelled nutrient and produced organic matter offshore. While a substantial portion of the offshore transported organic matter leaks into the interior of SCS and never comes back, the recirculation and quasi-stationary anticyclonic eddy trap the organic matters locally, and hinder further leakage of available nutrients in VBUS. The locally recirculated nutrient is then upwelled in the bottom Ekman layer, rejoining the production process over the shelf. (b) Non-separation state: During the non-separated circulation, the along-isobath circulation transports the organic matter northward. The**
**leakage of organic matter reduces the nutrient inventory in the VBUS. The loss of nutrients diminishes the nutrient inventory available for remineralization and upwelling, further inducing a reduction in the production process.**