# Peer review of "Physical Modulation to the Biological Productivity in the Summer Vietnam Upwelling System"

_Ocean Science, 2018_

## Referee Comment (RC1) · Anonymous Referee #1 · 15 Apr 2018

This study showed a close spatio-temporal covariability between biogeochemical module based on CoSINE model and kinetic energy based on ocean current from ADT in summer Vietnam upwelling system. The model results show that weakened circulation and eddy activity, with ∼21% less nitrate inventory and ∼16% weaker primary productivity when separation current is absent.

My impression on this manuscript is that two different physical and biological perspectives are well linked together to address productivity changes in response to physical forces. However, I found some misleading facts from the results that this paper discussed. Thus, I would like to recommend to accept this paper for the publication of OS after some revision.

Major Comments Is there any critical condition to explain that the elevated kinetic energy and intensified circulation can be explained by the separation of the upwelling current system (as described in Abstract).

Authors discuss Figure 5 for model comparison with OISST and VGPM NPP. To me, Model SST and NPP are not quite similar. Although authors admit its discrepancies, authors need to estimate this SST differences can cause how much uncertainties to obtain the result from covariability in physical-biological interaction.

For Fig. 7 (Line 218), "Subsurface CHL maxima appears at ∼35 m, which is somewhat shallower than that in the observation." Authors also need to discuss this depth differences leading to how much uncertainties to obtain current results.

Minor Comments

Fig. 1, Magenta diamonds for observation stations are not clear. Change them with another better color (maybe black color)

Fig.2 shows CHL concentraton in Fig. 2a. I am wondering whether its magnitudes are right. It seems too low. Authors need to check.

---

## Author Comment (AC1) · 12 May 2018

The response was uploaded in the form of a supplement document.

Please also note the supplement to this comment:
https://www.ocean-sci-discuss.net/os-2018-9/os-2018-9-AC1-supplement.zip
* * *

---

## Referee Comment (RC2) · A. M. P. Santos (Referee) · 12 Jun 2018

[referee-annotated manuscript omitted]

---

## Referee Comment (RC3) · J. Dippner (Referee) · 21 Jun 2018

General Comments

Most of the scientific papers published about the South China Sea (SCS) are dealing with remote sensing or numerical modeling. Only few papers present in-situ observations, which are ignored by the authors. Unfortunately this is also one of the major problems within this paper, because a couple of relevant papers, which deal with this problem, were ignored. See reference list at the end.

The fact that higher wind speed causes a stronger upwelling and a higher nutrient flux into the euphotic zone, which is connected with higher primary production, is not new. It is text book knowledge. This process has been quantified in the SCS upwelling area

for normal and post El Nino years inside the upwelling area and offshore (Voss et al. 2006, Bombar et al. 2010). Hence the finding of the authors is not new. It is known since more than 10 years!

In the abstract the sentence "The elevated kinetic energy and intensified circulation can be explained by the separation of the upwelling system" is the same misinterpretation as in the Liu et al. (2002) paper. The opposite is true. The stronger monsoon intensifies the circulation. If the velocity reach a critical value a jet is detached from the coast. This is the classical Gulf Stream detachment problem discussed by Haidvogel et al. (1992) and Marshall & Tansley (2001). A similar detachment modulated by the ITCZ occurs in the SCS and is described by Dippner et al. (2013). Hence, this aspect is also not new.

The model application is not well posed and the validation is rather problematically. The discussion is a mixture of trivial statements and speculations. These aspects are outlined below. In addition, I have problems with the presentation. There are many Chinese references, however, I miss the fundamental theoretical papers on upwelling (see references) as well as the classical papers on upwelling observations, which are given e.g. in the references of the review by Mittelstaedt (1986).

To conclude: I cannot find any aspect, which merits publication. The paper is a mixture of textbook knowledge, physical misinterpretation, trivial statements, speculation and improper referencing. Therefore, I recommend the editor to reject the paper.

Specific Comments

The motivation of this paper, the so-called "contradictory conclusion", is funny. There is no contradiction. Both papers, the Hein et al. (2013) and the Liu et al. (2002), are correct. The conclusions in these papers were different, because different years were considered. The observations in the Hein paper were made in 2003, whereas the observation in the Liu paper were made in 1992, 1998 and 1999. The authors may have a look to the Multivariate ENSO Index. Drifter observations in 2003 indicated a lateral transport (Dippner et al. 2011) and the physical mechanisms behind the offshore

transport and the transport parallel to the coast were explained by Dippner et al. (2013). So, what is the scientific question of the paper and what are the hypotheses?

The model has a resolution of 1/10 degree. The width of the upwelling are is 42 km. That means that the upwelling area is resolved with less than 4 grid points. Such a resolution of the upwelling area is not sufficient for any conclusions on dynamical processes. Therefore, I recommend to remove the word "upwelling" from the title.

After spin-up, the model runs from 2002 to 2011 and the period 2005 to 2011 was analyzed. The seasonal signal was filtered and from the inter-annual variability composites of high and low chlorophyll were constructed. However, it is not clear how the "normal year", the "no advection" or the "El Niño" were constructed.

The model considers picoplankton, diatoms and coccolithiphorids as functional groups. These functional groups are not representative for the SCS. Dinoflagellates, Phaeocystis spp. and nitrogen fixing bacteria are not considered, although they play a major role in the SCS phytoplankton (Bombar et al. 2011, Doan-Nhu et al 2010, Loick-Wilde et al. 2017).

No information is given on initial conditions of the biogeochemical model. Without a sensitivity analysis, the statement that the ecosystem model is insensitive to initial conditions is not serious.

The authors used HNA and LNA as criteria for the construction of composites. This is rather problematic HNA and LNA are not robust variables. NPP is far away from any similarity with observations. Seasonal variability is much higher than inter-annual variability. There is no serious reason to use HNA- and LNA-composites Strong and weak monsoon would be much better criteria for the construction of composites.

The model validation is not convincing. In this context it is important to state that the 3D figures are not helpful. If the authors present an upwelling model, I would like to see a vertical cross section normal to the coast, which should indicate the upwelling of

the isopycnals and the poleward undercurrent, which is a quality criterion of upwelling models (O'Brien & Hurlburt 1972).

The model is $\sim 1°$C to cold, the modeled NPP does not fit and the estimated kinetic energy is too high. Nevertheless, the authors try to convince the reader on the reasonable well agreement. Furthermore, it should be clearly mentioned that the biannual signals were transient signals, which were not present every year. The authors mentioned that the reasons for discrepancies in validation is insufficient horizontal resolution, unrealistic parameterization etc., but these shortcomings are accepted. Why don't they use a model with a sufficient horizontal resolution, realistic parameterization etc.? Please explain.

The biogeochemical model produced results far away from reality. From observations, it is known that strong blooms in the upwelling area can be addressed to strong monsoon due to a northern position of ITCZ (Dippner et al. 2013), which causes a specific distribution of characteristic water masses (Dippner & Loick Wilde, 2011) and their corresponding specific species distributions (Loick-Wilde et al. 2017). In contrast, in the oligotrophic offshore area, production can be directly addressed to nitrogen fixing bacteria (Bombar et al. 2010, 2011).

L247: The statement "Part of the ammonium could then fuel nitrification and production . . ." is pure speculation. It is not shown.

The chapter Discussion has the character of a Results chapter. Normally in the discussion, new findings were discussed in the context of existing literature. This is not done. The paragraph on biogeochemical cycles should be skipped. The four mentioned cycles are either trivial or speculation in the sense of not shown. E.g., "upwelled water . . . stimulate high production" is a trivial statement.

The paragraph 4.2 is a collection of trivial statements. A comparison of two model runs with and without advection is not helpful in understanding dynamics.

The statement "the more intensive separation, the larger KE in VBUS, and vice versa" in not correct. KE is not a meaningful quantity because separation occurs if the velocity (not KE) reaches a critical value. The statement "high KE is linked to accelerated biogeochemical cycle" is speculation, it is not shown. What means in this context "accelerated". I don't believe that KE has an influence on biological turn-around times.

The conclusion has the character of a summary. It is a repetition of previous speculations.

The statement "numerical experiment was designed to reproduce the non-separated circulation pattern, while maintaining the external monsoon forcing" documents not well posed modelling. From literature it is known that the intensity of monsoon and the connected inter-annual variability in ITCZ are responsible for the fine structure in the Vietnamese upwelling area.

Technical Comments

The ms has too much acronyms

The reference Dippner et al. (2006) was published in 2007.

Equation 1 goes back to Ekman (1905), to whom belongs the credit and not Chen et al. (2012) or Gruber et. (2011).

No information on the drag coefficient is given.

Sloppy formulation: "near-surface geostrophic current". Skip the word geostrophic.

What is the reference level (layer of no motion) of the dynamic topography? Please explain.

The Statement "nonlinear advection is important to the separation of the coastal jet" should not been addressed to Gan and Qu (2008) or Wand et al. (2006). The credit belongs to Haidvogel et al. (1992) and Marshall & Tansley (2001).

L 165 wrong dimension, should read m2s-2.

I cannot see a magenta box.

L202 "the physical and biological parameters" is a wrong formulation. Parameters should be replaced by variable, because a parameter is a quantity, which cannot be measured and must therefore be parameterized, as the name says.

L208 Contradiction: Why ageostrophic components contribute to the kinetic energy? This is not compatible with the definition of kinetic energy. Please explain.

L220 Why a lag suggests a significant regulation of physical forcing? Please explain.

L233 What means "the current dissipates freshwater"? Please explain.

The figures are hard to read (too small legends or axes labeling) and not very informative. The main reason is the perspective view, which is surely nice to see, but the essential information remains hidden.

References Bombar, D. et al. (2010) J Geophys Res, 115, C06018 Bombar, D. et al. (2011) Mar Ecol Prog Ser, 424, 39-52 Dippner, J.W. & Loick-Wilde, N. (2011) J Mar Syst, 84, 42-47 Dippner, J.W. et al. (2011) Harmful Algae, 10, 606-611 Dippner, J.W. et al. (2013) J Geophys Res. Ocean, 118, 1618-1623 Doan-Nhu, H. et al. (2010) J Mar Syst, 83, 253-261 Ekman, V.W. (1905) Ark. Mat. Astr. Fys. 17(26) 74pp Ekman, V.W. (1923) Ark. Mat. Astr. Fys. 2, 1-52 Haidvogel, D.B. et al. (1992) J Phys Oceanogr, 22, 882-902 Loick-Wilde, N. et al. (2017) Progr Oceanogr, 153, 1-15 Marshall, D.P., Tansley, C.E. (2001) J Phys Oceanogr, 31, 814-837 Mittelstaedt, E. (1986) Landolt-Börnstein New Series V/3c, 135-166 O'Brien, J.J. & Hurlburt, H.E. (1972) J Phys Oceanogr, 2, 14-26 Stommel, H. (1956) Deep Sea Res. 3(4) 273-278 Voss, M. et al. (2006) Geophys Res Lett, 33, L07604 Yoshida, K. (1967) Jpn J Geophys 4, 1-75 Yoshida, K. & Mao H.L. (1957) J Mar Res, 16, 40-53

Joachim W. Dippner

---

## Author Comment (AC2) · 19 Aug 2018

**Comments from reviewer#2**
**The authors present a study of the physical-biological coupling of the coastal upwelling off Vietnam. They use in situ, remote sensing and model data. The model is validate with the in situ and remote sensing data, which gave some confident to its outputs. The MS is well written and present interesting results, concluding with a schematic interpretation of the processes related to the variability of primary productivity. Thus, it is my opinion that the MS should be accepted after minor revision taking into the account the comments I made directly in the PDF in annex.**

Response: We thank the positive comments and careful editing from the reviewer. Following the comments in the pdf file, we revised the manuscript as listed below.

Line 19: Yes, separation in the manuscript indicates alongshore current separates from the coast. Modified.

Line 22: We use the depth-integrated nitrate concentration, which is the nutrient inventory here. The unit is mmol m$^{-2}$.

Line 27 : Figure 1 revised as follows. We added several labels of the geographic locations.

[Figure]

[Figure]

Line 32: Thank you for providing these references. Cushing (1969) and Bakun (1996) were added to the citation list.

Line 94: Revised. Now this sentence read: "We use the upwelling intensity (UI) **or the "Bakun index" (Bakun, 1973)** as a proxy to measure the strength of upwelling (Chen et al., 2012; Gruber et al., 2011), following the classical paper of Ekman (1905)…"

Line 108&109 (now in Line 110 and 114): Agree and corrected.

Line 171 (now in Line 181): Agree. This sentence was modified as: "**KE and NPP both present biannually signals in most years**, i.e., peaks in summer and winter, as well as complex non-seasonal signals"

Line 173 (now in Line 183): here "$p < 0.01$" was added.

Line 177 (now in Line 185): Notice that we removed this sentence and replaced it with more relevant discussion.

Line 232 (Line 253): Corrected as "these shortcomings **could be** accepted."

Line 250 (Line 273): Corrected.

Line 315 (Line 358): Now it reads: "As **presented in** the schematic diagram in Fig. 12,"

Line 330: We included the abbreviation in appendix A following Sect. 6. We also removed some unnecessary acronyms to increase the readability.

Fig. 2: The scales for two subplots were the same. However, noting that in Fig. 2a the

nearshore CHL can be as large as 5.0 mg m$^{-3}$ exceeding the color scale, we replaced the color scale with a log2 scale as follows. We also revised the caption following your suggestions. Thank you.

[Figure]

Figure 2 (a) Summertime (MJJAS) average of surface chlorophyll concentration (color shading, unit: mg m$^{-3}$) from MODIS, overlapped white contours are mean ADT with the arrows showing the directions of geostrophic currents. Gray box is the region of interest (VBUS), while AE shows the center of the anticyclone. (b) **The region of interest:** standard derivation of surface chlorophyll (color shading, unit: mg m$^{-3}$) overlaid with the contours of surface KE with an interval of 0.1 from 0.1 to 1.0 (unit: m2 s$^{-2}$). **The magenta dot-dash contour delimits the ocean region over which chlorophyll and KE are averaged (see text)**.

---

## Author Comment (AC4) · 19 Aug 2018

**We thank the reviewer for the comments, which improve our manuscript. Our responses are in blue.**

**General Comments**

**1. Most of the scientific papers published about the South China Sea (SCS) are dealing with remote sensing or numerical modeling. Only few papers present in-situ observations, which are ignored by the authors. Unfortunately this is also one of the major problems within this paper, because a couple of relevant papers, which deal with this problem, were ignored. See reference list at the end.**

We accept this criticism and have included the additional literature provided by the reviewer and have (briefly) described their relevance to our study.

**2. The fact that higher wind speed causes a stronger upwelling and a higher nutrient flux into the euphotic zone, which is connected with higher primary production, is not new. It is text book knowledge. This process has been quantified in the SCS upwelling area for normal and post El Nino years inside the upwelling area and offshore (Voss et al. 2006, Bombar et al. 2010). Hence the finding of the authors is not new. It is known since more than 10 years!**

(See also related response to Comment#4 below).

In this response, we want to clearly define what is meant by "text book knowledge" of wind-induced upwelling: it is direct wind-induced upwelling in the ("text-book's") sense of "quasi two-dimensionality" of a cross-shore-vertical section coastal ocean upwelling driven by along-shore wind. The definition includes a coast, along-shore wind with scale that is much larger than the baroclinic radius of deformation (i.e. basically spatially uniform wind), and a rotating ocean basin which may have a sloping shelf (e.g. Gill's book, 1982). In the following, this will henceforth be referred to as 'text-book upwelling'.

We agree that the direct wind-induced upwelling and its potential connection with higher primary production is well studied. However, the real world of biophysical inter-connection is (fortunately) far richer than what we might have learnt in 'text books.'

Recent examples of productivity not directly wind-induced (in the text-book's sense) in boundary currents are: Nguyen et al. (2015) and Oey et al. (2018). In these works, it is clear that the text-book upwelling dynamics has little connection with productivity. In the Vietnam upwelling region, Voss et al. (2006), for example, also noted that the Mekong River may explain the much higher nitrogen fixation outside the upwelling strip of the coast, presumably in part due to increased stratification caused by the spreading of lower surface salinity in the river plume [e.g. Oey and Mellor 1993]. Voss et al. (2006) did not

dynamically demonstrate that increased stratification was the cause, but we tend to agree with them, as we explained dynamically in Huang and Oey (2015) and Lin and Oey (2016). It is clear that currents and dynamics other than the text-book upwelling circulation play a role. Our model in the present study is based on the primitive equation with thermodynamics; 'circulation' here therefore includes also effects of stratification (i.e. baroclinicity). It is not obvious how one might *clearly* separate the different effects, as the strength of the text-book upwelling is most likely linked to other processes.

In our study, we found that, in addition to text-book upwelling, the kinetic energy (KE) of the current and anticyclone (i.e., the "circulation") also influence productivity.

In the manuscript, we show and explain (see Table S1 below) the $R^2$ between the net integrated primary production (NPP) and (a) the upwelling index UI (i.e. text-book upwelling, related to the along-shore wind stress) and (b) the kinetic energy KE of the circulation; the KE is used as a proxy for the strength of the circulation (referred to as simply "circulation"). Table S1 shows that the UI accounts for 45% of the NPP-variance, while KE accounts for 49%, i.e. the KE can explain as much, and actually slightly more, NPP variance than UI. However, the UI and KE are not independent. The KE is in part wind-forced - while the regression analysis cannot tell us cause and effect, it is reasonable in our case to assume this. Thus UI (through the wind stress) accounts for 32% of the KE variance. It is important to realize that this does *not* mean that the *uniform* alongshore wind stress that causes the text-book upwelling can actually explain 32% of the KE variance. This is because the regional wind stress off Vietnam cannot be separated from the large scale wind stress curl that is crucial to the current separation – i.e. the circulation. In other words, text-book upwelling can *at most* explain 32% of the KE variance, most likely less than 32%.

The simple analysis in Table S1 suggests that (1) text-book upwelling alone is unable to explain the full variability of NPP off Vietnam; (2) circulation plays an important role in contributing to the NPP variability off Vietnam; and (3) a large part of the circulation (at least ~68%) is unexplained by the *uniform* regional wind alone off the Vietnamese coast.

Table S1 R-squared[#] among variables average within VBUS region

| X \ Y | NPP | UI | KE |
|-------|-----|-----|-----|
| UI | 0.4548 | - | 0.3240 |
| KE | 0.4930 | 0.3240 | - |
| UI & KE | 0.6046 | - | - |

**p<0.01 for all correlations. NPP: Net integrated primary production; UI: Upwelling intensity; KE: Kinetic Energy.**

The main goal of our study is to understand and explain the role of the circulation in contributing to the NPP variance. We found, and quantitatively demonstrated using model

experiments, that the boundary current separation plays an important role. We therefore agree with the reviewer about the important role of current separation, but fundamentally disagree with the Dippner et al. (2013) interpretations, who as explained below (in Comment 4) mis-applied the Marshall and Tansley (2001) formula.

**3. In the abstract the sentence "The elevated kinetic energy and intensified circulation can be explained by the separation of the upwelling system" is the same misinterpretation as in the Liu et al. (2002) paper. The opposite is true. The stronger monsoon intensifies the circulation. If the velocity reach a critical value a jet is detached from the coast.**

We agree that this sentence may be misleading. Here we did not attempt to emphasize causality relation. Now the modified sentence reads:

"Results from a physical-biological coupled model reveal that the elevated kinetic energy **is linked to** the strength of the current separation from the coast."

We added the following discussion as an appendix:
"Qualitatively, both the analysis based on remote sensing data and model results suggest the separation flow is linked with stronger KE (~65% larger in HNA case then LNA case, Sect. 3.1). Moreover, a separation index is defined to quantitatively explain the relation between the flow separation and intensified circulation. The separation index (SI) can be written as:

$$SI = \sum \frac{u \cdot \cos \varphi + v \cdot \sin \varphi}{\sqrt{u^2 + v^2}}, \tag{S1}$$

where $u$ and $v$ are the two surface velocity components, and $\varphi$ is the angle between the topography gradient and the positive $x$ axis. This SI is essentially the area-averaged cross-isobath velocity normalized by the magnitude of the velocity, which is used to quantify flow separation here.

Fig. S1 shows the distribution of SI in Aug 2010. Positive values indicate that the flow is separating and downslope, and that may be seen off Vietnam south of the coastline bend. Large SI (~1.0) can be observed near the separation point ~11.5°N. Taking spatial average over the box region in Fig. S1, there is a good positive correlation (R=0.7175, p<0.01) between log(KE) and SI (see Fig. S2). Moreover, SI may be seen to generally increase with KE to a value of 0.25~0.3 and then it levels off (i.e. the slope becomes less) – see the red and blue lines in Fig. S2. The log(KE) and SI thus appears to show a logistic-type behavior, in which SI asymptotically approaches some maximum value (in this case ~0.3). This suggest that the strong flow separation and elevated KE are tightly linked. From Fig. S2, the value of KE ≈ 0.1 m²s⁻² appears to be a critical value.

Dynamically, the nonlinear advection term in the momentum equation can be written as the vector invariant form [see e.g., Gill (1982)]:

$$\vec{u} \cdot \nabla \vec{u} = (\nabla \times \vec{u}) \times \vec{u} + \nabla(\frac{1}{2}|\vec{u}|^2)$$

This decomposition directly links the nonlinear advection term and the gradient of KE (which scales KE over a length scale L). Meanwhile, the nonlinear advection is an important mechanism in driving flow separation [see, for instance, Oey et al. (2014)]. Stronger advection suggests intense cross isobath flow. Therefore, a dynamic linkage between the flow separation and the intensified KE and circulation can also be established, further supporting this argument.

[Figure]

Figure S1 Example of modeled SI in Aug 2010.

[Figure]

Figure S2 Summer-month (MJJAS) KE vs SI averaged over the box region in Fig. S2 (overall R=0.7175).

(*End of Appendix A)

We respectfully disagree with the reviewer's last 2 sentences: "**The stronger monsoon intensifies the circulation. If the velocity reach a critical value a jet is detached from the coast.**" These are based on Dippner et al. (2013). Taken together, and the Reviewer comment#2 above, the reviewer seems to suggest that text-book upwelling intensifies and causes jet separation from the coast. As we point out below, Dippener et al. (2013) mis-applied the Marshall and Tansley (2001) formula.

**4. This is the classical Gulf Stream detachment problem discussed by Haidvogel et al. (1992) and Marshall & Tansley (2001). A similar detachment modulated by the ITCZ occurs in the SCS and is described by Dippner et al. (2013). Hence, this aspect is also not new.**

Our paper does not deal with western boundary current (WBC) separation, and we checked through the manuscript again to make sure that no such claim is made.

On the other hand, the reviewer's comments are based on the WBC separation issue and the work of Dippner et al (2013), and there is a need for us to respond.

To the best of our knowledge, the problem of separation of a WBC, the Gulf Stream included, remains unresolved to this date ; the following is from Chassignet and Marshall (2008):

*"...Identifying the dynamics responsible for western boundary current separation has been a long-standing challenge. It is fair to say that a proper western boundary current separation ... is the result of many contributing factors.. There is yet no single recipe that would guarantee a correct separation ..."*

Neither Haidvoget et al. (1992) nor Marshall & Tansley (2001) claimed that they solved the WBC separation problem. The authors recognized that the problem is complex and depends on various factors; for examples: wind stress curl, topographic effects including coastline geometry and bottom topography, inertia & nonlinearity, deep WBC (e.g. the Gulf Stream), unresolved eddies (and how we may parameterize them, as well as the seemingly simple questions of free-slip or no-slip BCs), PV crisis, adverse pressure gradient, boundary current collision, outcropping of isopycnals, multiple equilibria, eddy-topography interaction, surface cooling etc. On reading Dippner et al. (2013), we failed to see how they have dynamically demonstrated the VBUS separation problem. There is a misunderstanding of what a WBC separation is; in particular the authors mis-understood and mis-applied the Marshall and Tansley (2001; MT2001) formula (in particular, their equation 11) to the South China Sea.

First, the MT2001 theory is applicable only in steady state, which is not the case of the VBUS forced by the monthly-varying monsoon. In applying the formula, Dippner et al (2013) therefore implicitly assumed steady state. However, they did not check the validity of this assumption. A lower bound for this time can be estimated based on the time it

takes for a long baroclinic wave to propagate across the basin [Lighthill 1969 Proc Roy Soc Lond; Anderson and Gill 1975 DSR], which for SCS is ~ 1,000 km/$\beta R_d^2$. The mode-1 baroclinic radius of deformation $R_d \approx 80$ km near the separation latitude (Fig. S3 below). Thus the required time (lower bound) is ~ 90 days. The summertime VBUS WBC is likely to be continually varying and quite unsteady under the monthly varying monsoon wind (Fig. S4 below).

[Figure]

Figure S3 Baroclinic Rossby radius of deformation $R_d$ based on the WOA June climatology [see e.g. Xu and Oey JPO 2015].

[Figure]

Figure S4 South China Sea monthly climatological wind (m/s; vector scale in top left) and wind stress curl ($\times 10^{-7}$ N/m$^3$; color shading) based on the CCMP wind data from 1988 to 2009), for the month of (from left to right) May, June, July, August and September (same months as used in Dippner et al. 2013), showing significant monthly variation both in wind and wind stress curl.

Second, even if we assume that quasi-steadiness has been reached, the form of Marshall and Tansley's formula used by Dippner et al. is valid only for a coastline with a vertical wall (no shelf, no slope). This assumption is clearly invalid for Vietnam coast (see Fig. S5

below) characterized with a shelf/slope that narrows to the 'cape' (i.e., separation point), which further more is where/when such a topography convergence favors separation - the middle term of equation 8 in Marshall and Tansley (2001) neglected by Dippner et al.

[Figure]

Figure S5 Bottom topography (in meters) of the SCS, from Fig.1a of the manuscript.

Third, let us further relax the assumptions and assume that the vertical-wall coastline is appropriate. For the Gulf Stream, Marshall and Tansley used the 200-m isobath – firstly because it is a good proxy for the path of the Gulf Stream (Figure below), and secondly because their theory is inviscid and is not intended to apply over the shallow shelves, where bottom friction can be strong.

[Figure]

Figure S6 The Gulf Stream region. White contours show the 50 and 200-m isobaths. Color shading and black contours show the 16-year mean (1993-2008) sea-surface height (m) from satellite altimetry (AVISO). Note that the path of the Gulf Stream follows closely the 200-m isobaths up to the separation point where the isobaths makes a sharp turn to the north, as assumed (correctly) by MT2001. [From Xu and Oey 2011, JPO].

Using the 200-m isobaths, MT2001 obtained a radius of curvature $r \sim 200$ km. For the SCS/VBUS region (see previous Figure above), the 200-m isobaths runs nearly straight from south to north pass the Vietnamese *coastline* bend near 15ºN, yet Dippner et al. used $r = 114$ km – a curvature which is even sharper than that used for the Gulf Stream – which is clearly inappropriate.

These misunderstandings and mis-application of Marshall and Tansley (2001) lead to a false interpretation of the separation dynamics – e.g. Fig.6 of Dippner et al. (2013).

ENSO events & ITCZ analyzed in Dippner et al.:

The authors suggested that in post non-El Nino (i.e. "normal" year) 2004, "…The upwelling and the offshore transport of WM2 and OSW resulted in a blocking of the northward transport of MKGTW and caused the mentioned current separation…" This was contrasted with a post El Nino year 2003 without such 'blocking.' Authors alluded the difference to the intensity of upwelling (which we assume here is the "text-book" type mentioned by the reviewer in Comment#2 above) by the stronger SW monsoon wind in 2004, and then proceeded to apply the Marshall and Tansley (2001) formula (inappropriately as we have already shown above). The ITCZ conclusion was based on this "intensity of upwelling hypothesis" which we now examine.

While we agree that upwelling contributes to increased productivity etc (See response to Comment#2 of the role of 'circulation'), wind-driven upwelling current does not lead to current separation, no matter its strength. We show this using idealized model experiments to isolate physics.

A South China Sea domain 99-122E & 0-25N is used, closed on all 4 lateral boundaries to ensure a simple zero inflow/outflow boundary condition. In particular the influence of Kuroshio intrusion variability through the Luzon Strait (Chang and Oey 2012 JClim; Xu and Oey 2014 OD; Xu and Oey 2015 JPO; Lin et al. 2016 JPO) is eliminated to focus on wind-driven responses. The initial temperature is a function of "z" only: $T(z)=2+24*\exp(0.0018*z)$, which approximates the observed climatologically mean vertical profile in South China Sea (see e.g. Lin et al. 2016 JPO), the salinity is set constant = 35 psu (and remains so through the integration as no rivers nor surface fluxes were specified), and the ocean is at rest. At the surface, all fluxes are nil except the momentum flux i.e. wind stress. A seasonally-varying wind climatology from the long-term CCMP data was used to force the model. For each of the 4 experiments below, the model was integrated for 3 years, and after allowing 2-year spin up for the solution to reach quasi-equilibrium (see Xu and Oey 2015 JPO, and references cited therein), the third year output was analyzed. The four experiments are now described.

Exp.NoCurlNoTopo: The basin depth was set constant at 4000 m (i.e. no topographic variation), and the zonal and meridional climatological wind components were averaged meridionally and zonally respectively to eliminate wind curl (i.e. no curl; Figure S7

below, left panel). Thus the coast has a vertical wall and under a summertime southwesterly wind off the southeastern coast of Vietnam (8N~12N), this experiment approximately simulates the 'classical' wind-driven upwelling process (Gill, 1982, his book): offshore surface Ekman transport compensated (approximately in this case) by onshore bottom Ekman transport, and up-lifted isopycnals near the coast where cooler water surfaces, balanced through thermal wind by an along-shore coastal jet that is strongest near the surface. The jet's width is approximately the mode-1 baroclinic radius of deformation $R_d \approx 80$ km (see Figure given previously, or Xu and Oey JPO 2015). The cool surface water shows up as a thin strip of lower SST ($< 27.5$ °C; middle panel) that clings close to the Vietnamese coast, from southeastern coastline around the bend to about 16N latitude. In the 3-D simulation, the coastal jet becomes unstable (Durski & Allen 2005 JPO), as seen in the wavy structure of the 27.5 °C isotherm and the meandering current with offshore jets and spun eddies (right panel). Importantly, the sea-surface height SSH is small i.e. the surface is nearly everywhere flat (middle). There is no WBC and obviously one *cannot* apply the Marshall & Tansley formula. Except for the above-mentioned eddying flow from the unstable coastal jet, there is no boundary current separation.

[Figure]

Figure S7 The idealized experiment Exp.NoCurlNoTopo: constant-depth ocean forced by spatially uniform wind without curl. Shown are July mean fields: (left) wind vectors and wind stress curl ($\times 10^{-7}$ Pa/m, color shading); (middle) SSH anomaly (m; color shading) and SST (°C, showing only those <27.5; contours); and (right) surface to 50 m averaged currents (vectors with speeds < 0.1 m/s omitted for clarity).

Exp.NoCurlWithTopo: The model basin now has realistic topography, but the wind is still spatially uniform without curl (Figure below). Over a sloping shelf, a prograde front (frontal and bottom slopes have the same sign, see e.g. Oey 2008, JPO) is stabilized by topography (Mysak 1977 JPO; Ikeda 1983 JPO). This may be seen in the current plot (right panel) and a comparison with the corresponding flat-bottom case discussed above; off the Vietnamese southeastern coast, the current now flows parallel to the coastline with little meanders. Friction is also increased over the shallow shelf and the water is more vertically mixed near coast; the prograding (upwelling) front is confined to the outer shelf and slope (Allen 1995 JPO) and along most of the southeastern coastline south of the bend, the near-coast SST is not as cool as for the flat topography experiment Exp.NoCurlNoTopo, shown previously. As a result, the coolest water now is advected toward and accumulates near the bend, where isobaths rapidly converge. The SSH again shows weak signal south of ~12N,

no WBC, and no boundary current separation. [There are eddies in the northern half of the basin due to some kind of dynamical instability (e.g. Oey 2008 JPO; see also Xu & Oey 2015 JPO; and Lin et al. 2016 JPO) which, while they may be interesting, has little relevance to the separation issue being discussed.]

[Figure]

Figure S8 The idealized experiment Exp.NoCurlWithTopo: ocean with realistic topography forced by spatially uniform wind without curl. Shown are July mean fields: (left) wind vectors and wind stress curl (×10$^{-7}$ Pa/m, color shading); (middle) SSH anomaly (m; color shading) and SST ($^o$C, showing only those <27.5; contours); and (right) surface to 50 m averaged currents (vectors with speeds < 0.1 m/s omitted for clarity); the blue contour shows the 200-m isobath.

Exp.WithCurlNoTopo: The model now is forced with the spatially non-uniform climatological wind stress but the basin depth is set constant at 4000 m (Figure below). Off Vietnam, the wind stress curl shows a dipolar pattern positive (negative) north (south) of the bend near 12N. The wind stress curl drives southward (northward) Sverdrup interior flow in the southern (northern) South China Sea, and a northward (southward) WBC along the Vietnam's coast (Pedlosky 1979, his book; or Gill 1982, his book), as can be clearly seen in the current plot (right panel). The SSH also forms a dipole, negative (positive) north (south) near the bend mirroring the wind stress curl dipole, as can be expected. In the southern half of the basin the SSH is visibly higher compared to the previous NoCurl experiments. Because of the seasonally varying wind forcing, both WBC systems are continually evolving, and are never in a steady state. Nonetheless, boundary current separation can be seen off the bend near 12N, and is clearly forced by the wind stress curl dipole, not by the upwelling current which we already show in Exp.NoCurlNoTopo above produces no such separation. Comparing the SST for the two cases, it is also clear that the offshore ejection of cool water near the bend is strongly controlled by the separation, forced by the wind stress curl dipole.

[Figure]

Exp.WithCurlWithTopo: The final experiment now has realistic topography and is forced by the spatially non-uniform climatological wind stress. With the sloping shelf and slope, the SSH-dipole, hence also the separating current near 12N, become significantly stronger (Figure below). As isopycnals are dynamically lifted near the coast, more cooler subsurface water is brought near the surface along the southeastern coast of Vietnam, and the ejected water in the separated current becomes cooler and more extensive. Note however that, unlike the above constant-depth experiment Exp.WithCurlNoTopo, the northward-flowing WBC off the southeastern Vietnamese coast can no longer be supported by the planetary beta alone. Topographic beta now becomes important due to the northwestward sloping shelf off the southeastern coast of Vietnam. Together with bottom friction, topographic beta now contributes in modifying the WBC, as can be seen by comparing the currents in the two experiments (Csanady 1978 JPO; Xu and Oey 2011 JPO).

[Figure]

Figure S10 The idealized experiment Exp.WithCurlWithTopo: ocean with realistic topography forced by wind with curl. Shown are July mean fields: (left) wind vectors and wind stress curl (×10⁻⁷ Pa/m, color shading); (middle) SSH anomaly (m; color shading) and SST (ºC, showing only those <27.5; contours); and (right) surface to 50 m averaged currents (vectors with speeds < 0.1 m/s omitted for clarity); the blue contour shows the 200-m isobath.

In summary, these experiments clearly demonstrate that the idea that boundary current separation off the Vietnamese coast is caused by the wind-driven upwelling current (Dippner et al. 2013) is incorrect. A WBC cannot be formed by a wind-driven upwelling current, and therefore the application of the Marshall and Tansley's formula to the wind-driven upwelling current cannot be correct. It follows that attempts to explain the current separation and productivity by the strengths of upwelling current – the ENSO variability explained in Dippner et al, cannot be correct. Indeed, our experiments show the importance of the wind stress curl dipole. This suggests that the curl, instead of the intensity, of the wind is major driving factor of the separation. The absent of cold-water during post-ENSO summer is a result of weak wind stress curl (Figure below) and weak separation (compare the above experiments:

Exp.WithCurlWithTopo and Exp.NoCurlWithTopo). This explains the co-occurrence of cold water core and current separation in most years, since they are both largely controlled by the wind stress curl.

[Figure]

Figure S11 Wind vectors and wind-stress curl during (unit: $\times 10^7$ Pa m$^{-1}$) in Jul of normal years (left) and post-El Nino years (right).

Our linear experiment, presented in the manuscript, also shows a very weak separation, suggesting that intrinsic (nonlinear) dynamics of the ocean is important.

Last but not least, the physical-biological coupling (i.e., how the current modulates the productivity) in the VBUS was also less clear, especially about the quantification and detailed process. This is the main focus and finding of this manuscript, as in the abstract: "Here we show a close spatio-temporal covariability between primary production and **kinetic energy**." "The separated current forms an eastward jet into the interior South China Sea, **and the associated southern recirculation traps nutrient and favors productivity."**

**5. The model application is not well posed and the validation is rather problematically. The discussion is a mixture of trivial statements and speculations. These aspects are outlined below. In addition, I have problems with the presentation. There are many Chinese references, however, I miss the fundamental theoretical papers on upwelling (see references) as well as the classical papers on upwelling observations, which are given e.g. in the references of the review by Mittelstaedt (1986).**

There are actually no references in the Chinese language in the citation list, we do not want to think you actually meant "the references with Chinese authors". We cited papers which studied this field in this region and every reference was published in major journals in English and is worthy of citing. As you suggests, we added more fundamental theoretical papers in the citation list.

**6. To conclude: I cannot find any aspect, which merits publication. The paper is a mixture of textbook knowledge, physical misinterpretation, trivial statements, speculation and improper referencing. Therefore, I recommend the editor to reject the paper.**

We strongly do not agree with these points. Again, it should be emphasized that the main focus of this study is the current system's influence on the productivity off the Vietnam coasts, instead of the upwelling strength directly from the monsoon wind and its interannual variability. As explained in Comment 2, the variation of the circulation intensity explains additional variability in the production, which cannot be explained by the 'textbook upwelling' alone. Furthermore, although the mechanisms of current separation were already investigated by previous studies, the quantification of its influence on the ecosystem has never been investigated before, and the detailed physical-biological coupling processes are yet unclear. We quantify the recirculation's (and nonlinear effects') role, and provide an underlying mechanism. These are new in this study. To be more specific, we modified the title to "The Modulation of Nonlinear Circulation to the Biological Productivity in the Summer Vietnam Upwelling System".

**Specific Comments**
**7. The motivation of this paper, the so-called "contradictory conclusion", is funny. There is no contradiction. Both papers, the Hein et al. (2013) and the Liu et al. (2002), are correct. The conclusions in these papers were different, because different years were considered. The observations in the Hein paper were made in 2003, whereas the observation in the Liu paper were made in 1992, 1998 and 1999. The authors may have a look to the Multivariate ENSO Index. Drifter observations in 2003 indicated a lateral transport (Dippner et al. 2011) and the physical mechanisms behind the offshore transport and the transport parallel to the coast were explained by Dippner et al. (2013). So, what is the scientific question of the paper and what are the hypotheses?**

Here we used 'contradictory' to indicate apparent contradiction between literatures emphasizing the wind-induced upwelling (Xie et al., 2003) and those highlighted circulation's role (Kuo et al., 2004; Liu et al., 2002). To avoid confusion, this sentence was modified as: "However, the contribution from the recirculation was seldom quantified and compared with that directly from the upwelling, which motivates us to revisit the VBUS ecosystem and its connection with circulation."

An *annual* regression analysis between the summer-averaged NPP and January Multivariate ENSO Index (MEI) following Xie et al. (2003) draws a similar conclusion to our *monthly* analysis in Table S1. About half ($R^2=0.49$) of the variability in NPP can be explained by the ENSO variability. Considering both MEI and KE concurrently, the $R^2$ is 0.74, i.e., 25% more variability in NPP is explained. In addition, KE and MEI are mostly independent ($R^2$ for KE vs. MEI is 0.22). These are consistent with our analysis in Comment 2, demonstrating that the ENSO is not the only player in the NPP variation.

Hence, the scientific question and hypotheses can be seen, as we emphasized, in Line 65: "While the importance of local current system to the VBUS biogeochemical system has

been noted in some previous studies (Dippner et al., 2007; Kuo et al., 2004; Liu et al., 2012; Xie et al., 2003), the detailed processes are unclear. To what extent is the ecosystem in VBUS modulated by local circulation? How does the recirculation modulate productivity? How much does the local circulation contribute to production?"

**8. The model has a resolution of 1/10 degree. The width of the upwelling are is 42 km. That means that the upwelling area is resolved with less than 4 grid points. Such a resolution of the upwelling area is not sufficient for any conclusions on dynamical processes. Therefore, I recommend to remove the word "upwelling" from the title.**

In this manuscript, instead of the fine structure or mechanism of the upwelling, we study the production attributed to the recirculation related to the separation. Therefore, the focus region of this study is not confined to the "actual" upwelling strip of ~40 km wide as indicated by the first baroclinic Rossby radius of deformation (Dippner et al., 2007; Voss et al., 2006), but extending to a broader offshore region of ~3 degree wide (Fig. 2b, magenta contour). This point is now clarified in Line 111.

**9. After spin-up, the model runs from 2002 to 2011 and the period 2005 to 2011 was analyzed. The seasonal signal was filtered and from the inter-annual variability composites of high and low chlorophyll were constructed. However, it is not clear how the "normal year", the "no advection" or the "El Niño" were constructed.**

In Fig. 11, El Niño year indicates the post-El Niño summer in 2010, while the normal year means other years from 2005 to 2010. No advection is the result from the experiment (Table 1). We added the information in the caption of Fig. 11.

**10. The model considers picoplankton, diatoms and coccolithiphorids as functional groups. These functional groups are not representative for the SCS. Dinoflagellates, Phaeocystis spp. and nitrogen fixing bacteria are not considered, although they play a major role in the SCS phytoplankton (Bombar et al. 2011, Doan-Nhu et al 2010, Loick-Wilde et al. 2017).**

Other plankton species can be important in the SCS. However, current understanding and observation data are insufficient to build a spatial and temporal simulation of these species. Further development of existing ecosystem model is required to simulate these groups, which is out of the scope of this study. This point was added to Line 146: "Other planktonic groups can be important in the ecosystem of SCS (Bombar et al., 2011; Doan-Nhu et al., 2010; Loick-Wilde et al., 2017). However, to keep the ecosystem model simple and computationally affordable, these groups are not considered in the CoSINE model."

**11. No information is given on initial conditions of the biogeochemical model. Without a sensitivity analysis, the statement that the ecosystem model is insensitive to initial conditions is not serious.**

As we showed in Line 152: "The initial distribution of nutrients and dissolved organic matter was also interpolated from the WOA climatological data. Small values were analytically assigned to other ecosystem variables since the ecosystem module was **less sensitive** to the initial conditions of these variables". Some studies related with ecosystem modeling (Lu et al., 2015; Wang et al., 2013) was less sensitive to the initial condition (except for nutrients), which are cited here in Line 155.

**12. The authors used HNA and LNA as criteria for the construction of composites. This is rather problematic HNA and LNA are not robust variables. NPP is far away from any similarity with observations. Seasonal variability is much higher than inter-annual variability. There is no serious reason to use HNA- and LNA-composites. Strong and weak monsoon would be much better criteria for the construction of composites.**

The motivation of this composite map is to examine the different **circulation pattern** when the production is high/low. This makes the choice of HNA and LNA natural. To show the robustness, composite with different thresholds (75%, 70%, and 60%) is also computed, which presents a similar contrast between HNA and LNA. This point was added to Line 193: "Different thresholds (60% and 70%) were also tested and very similar results can be seen."
In terms of NPP simulation, our model indeed shows some discrepancy, as we noted in the manuscript. However, firstly, the relation between wind, circulation and production can also be found based on model outputs (Line 244). Secondly, the contrast between the upwelling region and offshore region was captured. And the discrepancy may also be related to overestimation of retrieved NPP near the coast (Loisel et al., 2017). For both model and remote sensing NPP, seasonal variability is higher than interannual variability. These give us some confidence to further investigate the physical-biological coupling based on the model.

**13. The model validation is not convincing. In this context it is important to state that the 3D figures are not helpful. If the authors present an upwelling model, I would like to see a vertical cross section normal to the coast, which should indicate the upwelling of the isopycnals and the poleward undercurrent, which is a quality criterion of upwelling models (O'Brien & Hurlburt 1972).**

As noted in the response to Comment 2, the current system's role in the ecosystem is focused. Our consideration is that 3D figure is capable to show the recirculation of ecosystem variables (e.g., nitrate, organic matter) from different perspectives. The maps

of plane and section are shown below, which, for instance, clearly show the doming of isopycnals and northward undercurrent to the deep. This figure was included in the supplementary.

[Figure]

Figure S12 (a-c) Modeled sea level (black contour, CI=0.1 m) overlaid with (a) surface current (color: magnitude in m s$^{-1}$; vector: flow direction), (b) surface primary production (mg C m$^{-3}$ d$^{-1}$), and (c) particulate organic carbon (mmol C m$^{-3}$). (d-f) Sections along 10°N: meridional velocity (positive in solid contours and negative in dashed, CI=0.1 m s$^{-1}$. Thick contours indicate zero value) overlaid with (d) potential density anomaly, (e) nitrate concentration (mmol m$^{-3}$), and (f) ammonia concentration (mmol m$^{-3}$).

**14. The model is ~1°C too cold, the modeled NPP does not fit and the estimated kinetic energy is too high. Nevertheless, the authors try to convince the reader on the reasonable well agreement. Furthermore, it should be clearly mentioned that the biannual signals were transient signals, which were not present every year. The authors mentioned that the reasons for discrepancies in validation is insufficient horizontal resolution, unrealistic parameterization etc., but these shortcomings are accepted. Why don't they use a model with a sufficient horizontal resolution, realistic parameterization etc.? Please explain.**

We admit that the modeled and observed time-series did not match so well. However, as a process-oriented simulation, the model provides us an approach to investigate the underlying mechanism, considering the focus is to investigate the positive correlation between the productivity and the circulation, which was captured by the model.
We agree that the biannual signal is a transient signal. This sentence was modified as "(appears) in most years…".

The insufficient horizontal resolution is a hypothetical reason. Increasing resolution does not always improve model simulation [see e.g., Sandery and Sakov (2017)]. We hence removed this reason. Notice that all models are simplifications of the real ocean. It is impossible to take all processes into consideration and hence parameters cannot be perfect. Of course, we dream of a model simulation without any bias and if we knew how to further improve the model, we would definitely try our best to improve it.

**15. The biogeochemical model produced results far away from reality. From observations, it is known that strong blooms in the upwelling area can be addressed to strong monsoon due to a northern position of ITCZ (Dippner et al. 2013), which causes a specific distribution of characteristic water masses (Dippner & Loick Wilde, 2011) and their corresponding specific species distributions (Loick-Wilde et al. 2017). In contrast, in the oligotrophic offshore area, production can be directly addressed to nitrogen fixing bacteria (Bombar et al. 2010, 2011).**

As we emphasized, the focus area of this study is to a broader area offshore ~three-degree wide. The results from Bombar et al. (2010) suggested that the nitrogen fixation "was a significant nitrogen source", which was estimated "2-25% of diffusive nitrate fluxes", but not *all nitrogen source* in the offshore region. Considering the large uncertainty in this estimation, it is very likely that other processes are also playing. And our study showed one possible process, i.e., the nonlinear recirculation related to nutrient trapping, in playing. The conclusion of these paper does not conflict with the papers listed here.

**16. L247: The statement "Part of the ammonium could then fuel nitrification and production . . . " is pure speculation. It is not shown.**

The *f*-ratio listed in Table 1 was estimated ~0.6, which suggests that that the ammonium supports regeneration production. This point was added to discussion in Line 294.

**17. The chapter Discussion has the character of a Results chapter. Normally in the discussion, new findings were discussed in the context of existing literature. This is not done.**

Additional discussion associated with existing literatures were added. See the revised part in the manuscript (text in red).
E.g., in Line 279: "In the summer VBUS system, it is generally agreed that the wind's predominant role in controlling the variability in the production of VBUS, especially on the inter-annual scale (Dippner et al., 2013). This is also the case in our analysis where UI contributes ~45% of the total variability in production."
Line 285: "The separated current system was considered to transport high-chlorophyll water offshore (Xie et al., 2003). In the offshore region, the production appeared to be elevated (Bombar et al., 2010). However, the fate of the offshore nutrients was rarely investigated in literature."

Line 341 "To a larger scale, the recirculation current couples coastal upwelling and offshore region in major coastal upwelling systems, e.g., in the Canary basin (Pelegrí et al., 2005)."

**18. The paragraph on biogeochemical cycles should be skipped. The four mentioned cycles are either trivial or speculation in the sense of not shown. E.g., "upwelled water . . . stimulate high production" is a trivial statement.**

We attempt to emphasize that the cycle is important in understanding the recirculation's role in the ecosystem. The four stages of the cycle are supported by analyzing model results. Here we show the evidence one-by-one.
(1) Supported by various papers focus on VBUS [e.g., Dippner et al. (2007)].
(2) Supported by the tongue-like structure of high-chlorophyll water offshore (Fig. 2a and Fig. 9h).
(3) Comparing the high production and low production scenarios in Fig. 10i, where the high production case shows a clear returning flow with high organic matters (Fig. 10h). On the contrary, the low production case presents a tendency of northward transport (Fig. 10g).
(4) The subsurface maxima of ammonium can be seen from Fig. 9f. Bottom Ekman: see high nutrient in the bottom boundary layer in Fig. 9e, and also Gan et al. (2009). The offshore remineralization can be supported by the high oxygen consumption found off Vietnamese coasts (Jiao et al., 2014).
These evidences were added in the text.

**19. The paragraph 4.2 is a collection of trivial statements. A comparison of two model runs with and without advection is not helpful in understanding dynamics.**

In classical coastal upwelling theory, the role of the nonlinear term is seldom considered. However, in this manuscript, the recirculation related with nonlinear advection appeared to be important. In the derivation of Marshall and Tansley (2001), the nonlinear term is also involved in the governing equation (their equation 1). The experimental case without advection/nonlinear term can be considered as an extreme case where the KE and the separation are very weak with lower production, as we discussed in this section. The nonlinear advection term plays a key role in modulating the ocean circulation, especially in the boundary current systems. So no-advection experiments were designed in many simulation works [e.g. Gruber et al. (2011)].

**20. The statement "the more intensive separation, the larger KE in VBUS, and vice versa" in not correct. KE is not a meaningful quantity because separation occurs if the velocity (not KE) reaches a critical value. The statement "high KE is linked to accelerated biogeochemical cycle" is speculation, it is not shown. What means in this**

**context "accelerated". I don't believe that KE has an influence on biological turn-around times.**

Additional explanation about the linkage between KE and current separation was added in the Appendix, which shows a critical value of KE can also be used to explain the separation. Actually, considering that KE = $0.5*(u^2+v^2)$, these two arguments are equivalent. Notice that in Table S1 and Comment 2, the KE and separation could be both wind-driven, while KE also presents variability (~68% actually) which cannot directly be explained by the wind speed. We also modified this sentence to "The larger KE, the more intensive separation, and vice versa". (It was "The more intensive separation, the larger KE, and vice versa.")

About the second sentence, the "accelerated" is deleted to avoid the confusion. It is better to put it in: "Low KE reduces the recirculation of nutrients" (corrected in Line 340), which was shown in section 4.2.

**21. The conclusion has the character of a summary. It is a repetition of previous speculations.**

In the first and second paragraphs of the conclusion, we summarized what was done and what was found in the previous parts of the manuscript. As we show in previous responses, these are supported in our results.

The latter two parts are distinct from the previous ones. In the third paragraph, we illustrate the underlying processes to explain how the recirculation contribute to the production. And in the fourth paragraph, potential research that may be carried out in the future was proposed.

**22. The statement "numerical experiment was designed to reproduce the non-separated circulation pattern, while maintaining the external monsoon forcing" documents not well posed modelling. From literature it is known that the intensity of monsoon and the connected inter-annual variability in ITCZ are responsible for the fine structure in the Vietnamese upwelling area.**

Surely the wind's role is very important, but this does not mean that other factors play no role in the separation. Our simulation shows the importance of nonlinear terms, in agree with Wang et al. (2006) and Marshall and Tansley (2001). Also see the discussion in Comment 4. To be more specific, this sentence was revised as "Numerical experiment was also designed to reproduce the **weak**-separated circulation pattern **without the recirculation**, while …"

**Technical Comments**
**23. The ms has too much acronyms.**

After removing CHL, POC and NCEP, We found it difficult to further reduce acronyms. We provided a table of all abbreviations in this paper in Appendix A (as Table S2 below).

Table S2 Abbreviations

| Acronym | Definition | Acronym | Definition |
|---|---|---|---|
| SCS | South China Sea | NPP | vertical-integrated net primary production |
| VBUS | Vietnam Boundary Upwelling System | PP | primary production (as a function of depth) |
| CCMP | Cross-Calibrated Multi-Platform data | KE | kinetic energy |
| MODIS | Moderate Resolution Imaging Spectroradiometer data | TFOR | Taiwan Strait Nowcast\Forecast system |
| VGPM | chlorophyll-based Vertically Generalized Production Model | CoSINE | Carbon, Silicon, Nitrogen Ecosystem model |
| ADT | Absolute Dynamic Topography | NO_ADV | model experiment with no advection term in momentum equations |
| OISST | Optimum Interpolation Sea Surface Temperature | UI | upwelling intensity |
| HNA/LNA | high/low-NPP anomaly scenario | | |

**24. The reference Dippner et al. (2006) was published in 2007.**

Thanks, corrected.

**25. Equation 1 goes back to Ekman (1905), to whom belongs the credit and not Chen et al. (2012) or Gruber et. (2011).**

Chen et al. (2012) and Gruber et. (2011) directly applied this equation in studying upwelling. We agree that Ekman (1905) should also be cited here. This index is also known as Bakun Index. Now it goes: "We use the upwelling intensity (UI) or **the "Bakun index" (Bakun, 1973)** as a proxy to measure the strength of upwelling (Chen et al., 2012; Gruber et al., 2011), **following the classical paper of Ekman (1905)…**"

**26. No information on the drag coefficient is given.**

Accept. Now the sentence reads: "$C_D$ is the drag coefficient **(constant, $1.3\times10^{-3}$)**".

**27. Sloppy formulation: "near-surface geostrophic current". Skip the word geostrophic.**

Removed.

**28. What is the reference level (layer of no motion) of the dynamic topography?   Please explain.**

We use absolute dynamic topography (ADT) in this study, which is directly measured by altimetry with respect to the geoid. This point was added in Line 87: "Gridded monthly-mean Absolute Dynamic Topography (ADT) **with respect to the geoid** at 1/4° resolution was acquired."

**29. The Statement "nonlinear advection is important to the separation of the coastal jet" should not been addressed to Gan and Qu (2008) or Wand et al.   (2006). The credit belongs to Haidvogel et al. (1992) and Marshall & Tansley (2001).**

This statement was: "**For the Vietnam boundary upwelling system**, since nonlinear advection is important to the separation of the coastal jet (Gan and Qu, 2008; Wang et al., 2006)". As studies focus on VBUS, Gan and Qu (2008) and Wang et al. (2006) were cited here. We agree that the two classical papers should be cited here. Now it reads: "… is important to the separation of the coastal jet (Gan and Qu, 2008; Wang et al., 2006), **which is familiar in the Gulf Stream separation problem in Haidvogel et al. (1992) and Marshall and Tansley (2001),** …".

**30. L165 wrong dimension, should read m²s⁻².**

Thank you for your carefulness. Corrected.

**31. I cannot see a magenta box.**

The magenta dot-dash contour region in Fig. 2b. On the blue background, this color should be clear to see.

**32. L202 "the physical and biological parameters" is a wrong formulation.   Parameters should be replaced by variable, because a parameter is a quantity, which cannot be measured and must therefore be parameterized, as the name says.**

Agree. Modified as "the physical and biological **variables**".

**33. L208 Contradiction: Why ageostrophic components contribute to the kinetic energy? This is not compatible with the definition of kinetic energy. Please explain.**

Modeled surface velocity includes other components (e.g., from the wind). To avoid misleading, we modified this sentence to "The overestimated KE is partially contributed by the **Ekman** components in the **modeled** surface current."

**34. L220 Why a lag suggests a significant regulation of physical forcing? Please explain.**

This sentence reads "When NPP is lagged for one month, the correlation is 0.752 with a p-value of 0.0214, suggesting a significant regulation of the physical forcing to the productivity." A p-value of 0.0214 suggests a significant regulation. The lag may be associated with the response time of ecosystem with respect to the physical forcing.

**35. L233 What means "the current dissipates freshwater"? Please explain.**

This sentence was re-written as: "The current also **disperses** freshwater from the Mekong River,…"

**36. The figures are hard to read (too small legends or axes labeling) and not very informative. The main reason is the perspective view, which is surely nice to see, but the essential information remains hidden.**

See response to Comment 13. If possible, could you indicate which figure has too small legends or labeling?

References

Bakun, A.: Coastal upwelling indices, west coast of North America, 1946-71, US Dept. Commerce NOAA Tech. Rep. NMFS-SSRF, 671, 1-103, 1973.

Bombar, D., Dippner, J. W., Doan, H. N., Ngoc, L. N., Liskow, I., Loick-Wilde, N., and Voss, M.: Sources of new nitrogen in the Vietnamese upwelling region of the South China Sea, J. Geophys. Res., 115, 2010.

Bombar, D., Moisander, P. H., Dippner, J. W., Foster, R. A., Voss, M., Karfeld, B., and Zehr, J. P.: Distribution of diazotrophic microorganisms and nifH gene expression in the Mekong River plume during intermonsoon, Marine Ecology Progress Series, 424, 39-52, 2011.

Chassignet, E. P. and Marshall, D. P.: Gulf Stream separation in numerical ocean models, 177, 39-61, 2008.

Dippner, J. W., Bombar, D., Loick-Wilde, N., Voss, M., and Subramaniam, A.: Comment on "Current separation and upwelling over the southeast shelf of Vietnam in the South China Sea" by Chen et al, J. Geophys. Res. Oceans, 118, 1618-1623, 2013.

Dippner, J. W., Nguyen, K. V., Hein, H., Ohde, T., and Loick, N.: Monsoon-induced upwelling off the Vietnamese coast, Ocean Dyn., 57, 46-62, 2007.

Doan-Nhu, H., Lam, N.-N., and Dippner, J. W.: Development of Phaeocystis globosa blooms in the upwelling waters of the South Central coast of Viet Nam, J. Mar. Syst., 83, 253-261, 2010.

Ekman, V. W.: On the influence of the earth's rotation on ocean-currents, Arkiv. Mat. Astron. Fys., 211, 1-52, 1905.

Gan, J., Cheung, A., Guo, X., and Li, L.: Intensified upwelling over a widened shelf in the northeastern South China Sea, J. Geophys. Res. Oceans, 114, 2009.

Gan, J. and Qu, T.: Coastal jet separation and associated flow variability in the southwest South China Sea, Deep Sea Research Part I: Oceanographic Research Papers, 55, 1-19, 2008.

Gill, A. E.: Atmosphere-ocean dynamics, Academic press, 1982.

Gruber, N., Lachkar, Z., Frenzel, H., Marchesiello, P., Munnich, M., McWilliams, J. C., Nagai, T., and Plattner, G.-K.: Eddy-induced reduction of biological production in eastern boundary upwelling systems, Nature Geosci, 4, 787-792, 2011.

Haidvogel, D. B., McWilliams, J. C., and Gent, P. R.: Boundary Current Separation in a Quasigeostrophic, Eddy-resolving Ocean Circulation Model, J. Phys. Oceanogr., 22, 882-902, 1992.

Huang, S. M. and Oey, L. Y.: Right-side cooling and phytoplankton bloom in the wake of a tropical cyclone, J. Geophys. Res. Oceans, 120, 5735-5748, 2015.

Jiao, N., Zhang, Y., Zhou, K., Li, Q., Dai, M., Liu, J., Guo, J., and Huang, B.: Revisiting the CO2 "source" problem in upwelling areas: a comparative study on eddy upwellings in the South China Sea, Biogeosciences, 11, 2465-2475, 2014.

Kuo, N.-J., Zheng, Q., and Ho, C.-R.: Response of Vietnam coastal upwelling to the 1997–1998 ENSO event observed by multisensor data, Remote Sens. Environ., 89, 106-115, 2004.

Lin, Y. C. and Oey, L. Y.: Rainfall-enhanced blooming in typhoon wakes, Scientific reports, 6, 31310, 2016.

Liu, K.-K., Chao, S. Y., Shaw, P. T., Gong, G. C., Chen, C. C., and Tang, T. Y.: Monsoon-forced chlorophyll distribution and primary production in the South China Sea: observations and a numerical study, Deep-Sea Res. Part I-Oceanogr. Res. Pap., 49, 1387-1412, 2002.

Loick-Wilde, N., Bombar, D., Doan, H. N., Nguyen, L. N., Nguyen-Thi, A. M., Voss, M., and Dippner, J. W.: Microplankton biomass and diversity in the Vietnamese upwelling area during SW monsoon under normal conditions and after an ENSO event, Prog. Oceanogr., 153, 1-15, 2017.

Loisel, H., Vantrepotte, V., Ouillon, S., Ngoc, D. D., Herrmann, M., Tran, V., Mériaux, X., Dessailly, D., Jamet, C., Duhaut, T., Nguyen, H. H., and Van Nguyen, T.: Assessment and analysis of the chlorophyll-a concentration variability over the Vietnamese coastal waters from the MERIS ocean color sensor (2002–2012), Remote Sens. Environ., 190, 217-232, 2017.

Lu, W., Yan, X.-H., and Jiang, Y.: Winter bloom and associated upwelling northwest of the Luzon Island: A coupled physical-biological modeling approach, J. Geophys. Res. Oceans, 120, 533-546, 2015.

Marshall, D. P. and Tansley, C. E.: An Implicit Formula for Boundary Current Separation, J. Phys. Oceanogr., 31, 1633-1638, 2001.

Nguyen, H. M., Rountrey, A. N., Meeuwig, J. J., Coulson, P. G., Feng, M., Newman, S. J., Waite, A. M., Wakefield, C. B., and Meekan, M. G.: Growth of a deep-water, predatory fish is influenced by the productivity of a boundary current system, Scientific reports, 5, 9044, 2015.

Oey, L.-Y., Chang, Y.-L., Lin, Y.-C., Chang, M.-C., Varlamov, S., and Miyazawa, Y.: Cross flows in the Taiwan Strait in winter, J. Phys. Oceanogr., 44, 801-817, 2014.

Oey, L.-Y. and Mellor, G.: Subtidal variability of estuarine outflow, plume, and coastal current: A model study, J. Phys. Oceanogr., 23, 164-171, 1993.

Oey, L.-Y., Wang, J., and Lee, M.-A.: Fish catch is related to the fluctuations of a Western Boundary Current, J. Phys. Oceanogr., 48, 705-721, 2018.

Pelegrí, J. L., Arístegui, J., Cana, L., González-Dávila, M., Hernández-Guerra, A., Hernández-León,

S., Marrero-Díaz, A., Montero, M. F., Sangrà, P., and Santana-Casiano, M.: Coupling between the open ocean and the coastal upwelling region off northwest Africa: water recirculation and offshore pumping of organic matter, J. Mar. Syst., 54, 3-37, 2005.

Sandery, P. A. and Sakov, P.: Ocean forecasting of mesoscale features can deteriorate by increasing model resolution towards the submesoscale, Nat Commun, 8, 1566, 2017.

Voss, M., Bombar, D., Loick, N., and Dippner, J. W.: Riverine influence on nitrogen fixation in the upwelling region off Vietnam, South China Sea, Geophys. Res. Lett., 33, 2006.

Wang, G., Chen, D., and Su, J.: Generation and life cycle of the dipole in the South China Sea summer circulation, J. Geophys. Res., 111, 2006.

Wang, J., Hong, H., Jiang, Y., Chai, F., and Yan, X.-H.: Summer nitrogenous nutrient transport and its fate in the Taiwan Strait: A coupled physical-biological modeling approach, J. Geophys. Res. Oceans, 118, 4184-4200, 2013.

Xie, S. P., Xie, Q., Wang, D., and Liu, W. T.: Summer upwelling in the South China Sea and its role in regional climate variations, J. Geophys. Res. Oceans, 108, 2003.

---

## Author Comment (AC5) · 19 Aug 2018

Please see the revised manuscript in the supplementary.

Please also note the supplement to this comment:
https://www.ocean-sci-discuss.net/os-2018-9/os-2018-9-AC5-supplement.pdf

---

## Author Response (AR2)

**Table of Content**

**1. Response to the comments from editor**

1. Thanks for the resubmission, which is much improved. This is certainly also due to the useful comments and critiques by the referees, in particular referee #3. Below are some comments which I encourage you to take into account.

Response: We agree that the critiques from the reviewers are extremely helpful in improving the quality of the manuscript. We thank the efforts and time paid by the reviewers. We also want to thank you for your consideration. The comments are taken into full consideration. Please see the point-by-point responses below and corresponding revision in the manuscript (all in blue color).

2. There is one answer to referee #3 that is not clear to me. This needs additional clarification:

11. No information is given on initial conditions of the biogeochemical model. Without a sensitivity analysis, the statement that the ecosystem model is insensitive to initial conditions is not serious.

Your Answer: As we showed in Line 177: "The initial distribution of nutrients and dissolved organic matter was also interpolated from the WOA climatological data. Small values were analytically assigned to other ecosystem variables since the ecosystem module was less sensitive to the initial conditions of these variables. Some studies related with ecosystem modeling (Lu et al., 2015; Wang et al., 2013) was less sensitive to the initial condition (except for nutrients), which are cited here in Line 180.

Editor: It is still not clear to me what you've done. Why would you assign small values (if you don't know them; and the reason you give I do not understand)

**Response: Our first-round revision here was not clear. We wanted to claim the biogeochemical model is INDEED sensitive to the initial condition of nutrients and dissolved organic matters, but NOT so sensitive to the variables of detritus and planktons. This part was re-written as:**

**"Our previous studies related with ecosystem modeling in the China Seas (Lu et al., 2015; Wang et al., 2016; Wang et al., 2013) suggested that the ecosystem module was more sensitive to the initial value of nutrients and dissolved organic matters. For other variables (i.e., detritus and planktons), the model converged to similar states even when these variables were initialized differently. Hence, for these ecosystem variables, small values were assigned as in Table S1." This table has been added as a supplementary table.**

**Table S1 Initial values for some of the ecosystem variables**

| Name | Description | Initial value | Unit | Note |
|------|-------------|---------------|------|------|
| S1_N | Nitrogen for pico-phytoplankton | 0.04 | mmol N m$^{-3}$ | |
| S1_C | Carbon for pico-phytoplankton | 0.265 | mmol C m$^{-3}$ | S1_N/16*106 |
| S1CH | Chlorophyll for pico-phytoplankton | 0.06 | mg m$^{-3}$ | |
| S2_N | Nitrogen for diatom | 0.08 | mmol N m$^{-3}$ | |
| S2_C | Carbon for diatom | 0.53 | mmol C m$^{-3}$ | S2_N/16*106 |
| S2CH | Chlorophyll for diatom | 0.12 | mg m$^{-3}$ | |
| S3_N | Nitrogen for coccolithophorids | 0.04 | mmol N m$^{-3}$ | |
| S3_C | Carbon for coccolithophorids | 0.265 | mmol C m$^{-3}$ | S3_N/16*106 |
| S3CH | Chlorophyll for coccolithophorids | 0.06 | mg m$^{-3}$ | |
| Z1_N | Nitrogen for small zooplankton | 0.02 | mmol N m$^{-3}$ | |
| Z1_C | Carbon for small zooplankton | 0.1325 | mmol C m$^{-3}$ | Z1_N/16*106 |
| Z2_N | Nitrogen for meso-zooplankton | 0.02 | mmol N m$^{-3}$ | |
| Z2_C | Carbon for meso-zooplankton | 0.1325 | mmol C m$^{-3}$ | Z2_N/16*106 |
| DD_N | Detritus nitrogen | 0.02 | mmol m$^{-3}$ | |
| DD_C | Detritus carbon | 0.1325 | mmol m$^{-3}$ | DD_N/16*106 |
| BAC_ | Bacteria carbon | 0.01 | mmol C m$^{-3}$ | |
| DDCA | Detritus inorganic carbon | 0.01 | mmol m$^{-3}$ | |

| **DDSi** | Detritus silicate | 0.03 | mmol m$^{-3}$ |
| --- | --- | --- | --- |

3.  In the Data section (§2.1) some of the data used is presented, which is fine. However, in §2.3 Model Description, many additional data are mentioned. Please pull all data together in one section, §2.1.

Response: Sure. The description of NCEP and WOA data was moved to Sect. 2.1. The source of the river runoff data was also added to Sect. 2.1. Data availability information was also added to Sect. 6.

4.  L101 delete: geostrophic, as you wrote in the response that you would do (point 27 of referee 3)

Response: Thank you for your carefulness. Removed.

5.  L139-140 "Climatological river discharges from the Mekong River and other major rivers are included as point sources." What is the origin of these data?

Response: The river runoff data was adopted from Dai et al. (2009), which contains observation-based monthly freshwater runoff. We added the origin of the data to the data section (2.1).

Dai, A., Qian, T., Trenberth, K. E., and Milliman, J. D.: Changes in continental freshwater discharge from 1948 to 2004, J. Clim., 22, 2773-2792, 2009.

6.  L146-148 This was added as a response to referee #3 (point 10). What would be the important issue here is, how would these other phytoplankton species change the results of the model? The manuscript needs a statement about this issue.

Response: Agree. It should be noted that these three papers reporting N-fixing bacteria (i.e., "other species") here are from one single cruise, while few further evidences from literature suggested that the N-fixation bacteria accounted for a significant portion of the production. For this reason, following sentence was added to the text:
"Surely, adding more planktonic groups would better depict the complex relationship in the ecosystem. However, considering the functional groups chosen here were dominating species widely observed in the SCS [e.g., Ning et al. (2004)], adding more planktonic species is unlikely to radically change the spatiotemporal variation of the modeled ecosystem."

7.  L148-150 "The CoSINE model was successfully applied in the study on the primary production (Liu and Chai, 2009), mesoscale eddy and its impacts (Guo

et al., 2015), and the phytoplankton community structure (Ma et al., 2013, 2014) in the SCS." What does this mean? How successful was the model and in what?

**Response: What we want to say is this model was applied well in this region and was evaluated several times in different studies. We re-wrote this part to give a detailed application of CoSINE in the SCS:**
**"The CoSINE model was well applied in various studies of the SCS. Liu and Chai (2009) investigated the seasonal and interannual variability of the primary productivity of the SCS at a basin scale. The modeled structure (e.g., phytoplankton community) and function (e.g., biological pump) of the ecosystem appeared to respond to both climatic variations (Ma et al., 2013, 2014) and mesoscale eddies (Guo et al., 2015), both well-captured by CoSINE. By taking a SCS average, the modeled NPP time-series showed a strong correlation (R=0.84) when compared with satellite-derived production (Ma et al., 2014). Details of the CoSINE results in SCS can also be referred to Lu et al. (2018). "**

8. L154 "Small values were analytically assigned to other ecosystem variables …" Which variables are meant here?

**Response: The variables related with detritus and planktons (including phytoplankton and zooplankton) were analytically initialized as those in Table S1 above.**

9. The second paragraph of the Conclusions ("To further investigate…") is like part of a summary. I suggest to shorten it and integrate it into the third paragraph.

**Response: Accepted. We shortened this paragraph from**

*"To further investigate the linkage between the circulation and the ecosystem, a physical-biological coupled model was developed. The modeled results were validated favorably compared with the remote sensing and in-situ observation data. In particular, model reproduced the positive contribution from the circulation intensity to the productivity. A numerical experiment was also designed to reproduce the weak-separated circulation pattern without the recirculation, while maintaining the external monsoon forcing."*

**to**

*"A physical-biological coupled model was applied to investigate the positive contribution from the circulation intensity to the productivity. A numerical experiment was also designed to reproduce the weak-separated circulation pattern without the recirculation."*

**And combined this with following paragraph.**

Starting from next page, the manuscript with tracking changes is shown with new page numbers and line numbers.

[revised manuscript text omitted]